# Parvalbumin-expressing interneurons coordinate hippocampal network dynamics required for memory consolidation

Nicolette Ognjanovski[1], Samantha Schaeffer[1], Jiaxing Wu[2], Sima Mofakham[3], Daniel Maruyama[3], Michal Zochowski[2,3,4] & Sara J. Aton[1]

Activity in hippocampal area CA1 is essential for consolidating episodic memories, but it is unclear how CA1 activity patterns drive memory formation. We find that in the hours following single-trial contextual fear conditioning (CFC), fast-spiking interneurons (which typically express parvalbumin (PV)) show greater firing coherence with CA1 network oscillations. Post-CFC inhibition of PV + interneurons blocks fear memory consolidation. This effect is associated with loss of two network changes associated with normal consolidation: (1) augmented sleep-associated delta (0.5–4 Hz), theta (4–12 Hz) and ripple (150–250 Hz) oscillations; and (2) stabilization of CA1 neurons' functional connectivity patterns. Rhythmic activation of PV + interneurons increases CA1 network coherence and leads to a sustained increase in the strength and stability of functional connections between neurons. Our results suggest that immediately following learning, PV + interneurons drive CA1 oscillations and reactivation of CA1 ensembles, which directly promotes network plasticity and long-term memory formation.

[1] Department of Molecular, Cellular, and Developmental Biology, University of Michigan, Ann Arbor, Michigan 48109, USA. [2] Applied Physics Program, University of Michigan, Ann Arbor, Michigan 48109, USA. [3] Biophysics Program, University of Michigan, Ann Arbor, Michigan 48109, USA. [4] Department of Physics, University of Michigan, Ann Arbor, Michigan 48109, USA. Correspondence and requests for materials should be addressed to S.J.A. (email: saton@umich.edu).

Recent findings have highlighted the importance of hippocampal activity patterns in episodic memory encoding and retrieval. A significant body of evidence suggests that coordinated activation of an engram-specific population of hippocampal neurons is sufficient for subsequent recall[1–5]. However, much less is known about hippocampal network dynamics involved in memory consolidation, and how these dynamics are coordinated following learning. While network activity in hippocampal area CA1 is essential for long-term memory consolidation, it is unclear (1) how newly learned information affects the CA1 network and (2) how CA1 activity is coordinated to promote long-term storage of this information. One hypothesis is that following learning, hippocampal oscillations (for example, those occurring in sleep) drive reactivation of specific neuronal ensembles; such reactivation could drive network plasticity[6–9]. Numerous studies have demonstrated sequential replay of network activity patterns associated with prior experience[10,11]. Selective offline reactivation of neuronal populations activated during a learning experience has also been widely reported[12–16]. Out of the two phenomena, reactivation appears less constrained to specific behavioural paradigms (for example, hippocampal place cell activation while traversing a maze), and more importantly, occurs at a time (just following a new experience) when it could play a pivotal role in consolidation. However, while behavioural state-dependent hippocampal oscillations (such as sharp wave ripples; SPWRs) may support sequential replay, the mechanisms mediating reactivation are still completely unknown[17]. Critically, it is still unclear whether hippocampal oscillations, replay, or reactivation are required for memory consolidation following novel learning.

Individual PV + interneurons in CA1 innervate large numbers of neighbouring pyramidal neurons[18–20]. Thus from a connectomic standpoint, they are well-positioned both to coordinate hippocampal oscillations and to pattern neuronal ensemble activity. To test the hypothesis that PV + interneurons optimally pattern network activity in the context of memory formation, we characterized firing in the CA1 fast-spiking (FS) (typically PV + ) interneuron population during contextual fear memory (CFM) consolidation. We find that these neurons (and neighbouring pyramidal neurons) show selectively enhanced firing coherence with delta and theta oscillations in the hours immediately following single-trial contextual fear conditioning (CFC). We also assessed how experimental manipulation of CA1 PV + interneuron activity affects memory consolidation. Under normal conditions, CFC leads to subsequent increases in CA1 delta, theta and ripple oscillations during sleep. Critically, these sleep-associated changes are maximal over the first 6 h following conditioning, when sleep plays a critical role in CFM consolidation[21], and the degree to which they are enhanced predicts the success of CFM consolidation. Pharmacogenetic inhibition of CA1 PV + interneurons disrupts CFM consolidation, and also blocks post-CFC enhancements in CA1 network oscillations. We also find that PV + interneuron activity is required to stabilize functional connectivity patterns among CA1 neurons, which is a hallmark of normal CFM consolidation in the hours following training. Finally, rhythmic optogenetic activation of PV + interneurons is sufficient to generate coherent firing rhythms and stabilize communication patterns across the CA1 network. Greater network stability and stronger network connections remain in CA1 for hours after rhythmic PV + interneuron stimulation ends, suggesting that this manipulation induces long-lasting synaptic changes. On the basis of these results, we conclude that PV + interneurons play a critical, instructive role in coordinating CA1 network communication after a novel learning experience, which drives both long-term network plasticity and memory formation.

## Results

**CA1 neurons' firing coherence increases after learning.** To characterize the dynamic response of CA1 neurons to newly learned information, we continuously recorded the firing of FS interneurons and non-FS (presumptive principal) neurons in wild-type mice during CFM consolidation, using chronically implanted stereotrode arrays (Supplementary Figs 1 and 2). Following a 24-h baseline recording period (during which CA1 neuronal and local field potential (LFP) activity was recorded continuously), C57BL/6J mice underwent single-trial CFC or sham conditioning. Recording continued over the next 24 h to assess changes in neuronal and network activity associated with CFM consolidation (Fig. 1a,b). FS interneurons (identified based on characteristically narrow spike waveform) showed dramatically increased spike-field coherence in the first 6 h following CFC, but not following sham conditioning (Fig. 1c)[22,23]. Similarly, neighbouring principal neurons showed greater spike-field coherence following CFC, but not sham conditioning (Fig. 1d). These changes were evident for both delta- and theta-frequency oscillatory activity in CA1, across behavioural states (Fig. 1e), and were not due to changes in post-CFC sleep architecture (Supplementary Fig. 3). These data suggest that rhythmic synchrony of neuronal firing in the CA1 network is augmented during active memory consolidation. Neuronal firing was also altered during SPWR events following CFC (Fig. 1f,g). While no change in spike-field coherence was detected during ripple oscillations, following CFC there was an increase in both the overall frequency of SPWR events (Fig. 1f) and FS interneuron firing rates during SPWRs (Fig. 1g).

On the basis of these findings, we carried out a series of experiments to test two hypotheses: (1) that CA1 FS interneurons (which are typically PV + ) augment network rhythms following CFC; and (2) that PV + interneuron-mediated rhythms play an essential role in memory consolidation.

**Inhibition of CA1 PV + interneurons blocks CFM consolidation.** To test these hypotheses, we used an adeno-associated virus (AAV) vector to express the inhibitory receptor hM4Di-mCherry in a CRE recombinase-dependent manner in area CA1 of Pvalb-IRES-CRE transgenic mice (Fig. 2a). This allowed transient pharmacogenetic inhibition of CA1 PV + interneurons upon systemic administration of the hM4Di ligand clozapine-N-oxide (CNO)[21]. We confirmed inhibition by CNO in vivo by characterizing c-Fos expression in these neurons 90 min after single-trial CFC (followed immediately by CNO administration). For these studies, we expressed hM4Di in right-hemisphere CA1 and a control (mCherry) vector in left-hemisphere CA1 in each mouse (Fig. 2b), to compare the number of c-Fos-immunopositive PV + interneurons with versus without pharmacogenetic inhibition. Under these conditions, the proportion of c-Fos+, hM4Di-expressing neurons was significantly lower than the proportion of c-Fos+, mCherry-expressing neurons (Fig. 2c,d), consistent with CNO-mediated inhibition of PV + interneurons after CFC.

To assess the behavioural and hippocampal network effects of PV + interneuron inhibition, mice expressing either hM4Di or mCherry bilaterally in CA1 were implanted with CA1 stereotrode arrays (Supplementary Fig. 4). Following a 24-h baseline recording period (during which baseline neuronal and LFP activity was recorded), mice underwent single-trial CFC followed by CNO administration (Fig. 3a). CFM was assessed (as context-specific freezing) 24 h later, when mice were returned to the same environmental context. In hM4Di-expressing mice, CNO administration immediately following CFC-disrupted CFM consolidation (Fig. 3b). Effects of PV + interneuron inhibition

on memory were not due to changes in post-CFC sleep architecture (which was similar between treatment groups; Supplementary Fig. 6)[24,25]. Nor were these effects due to induction of hippocampal seizures (that is, ictal spiking was absent from CA1 LFPs and behaviour was unchanged following CNO administration) or gross dysregulation of neuronal firing rates in CA1 (Supplementary Fig. 8). To test whether CFM deficits were driven by hM4Di expression alone, the same hM4Di-expressing mice underwent single-trial CFC 2 weeks later in a second, dissimilar environmental context and were subsequently administered a vehicle (dimethylsulphoxide, DMSO). Under these conditions, CFM consolidation in hM4Di-expressing mice was comparable to that

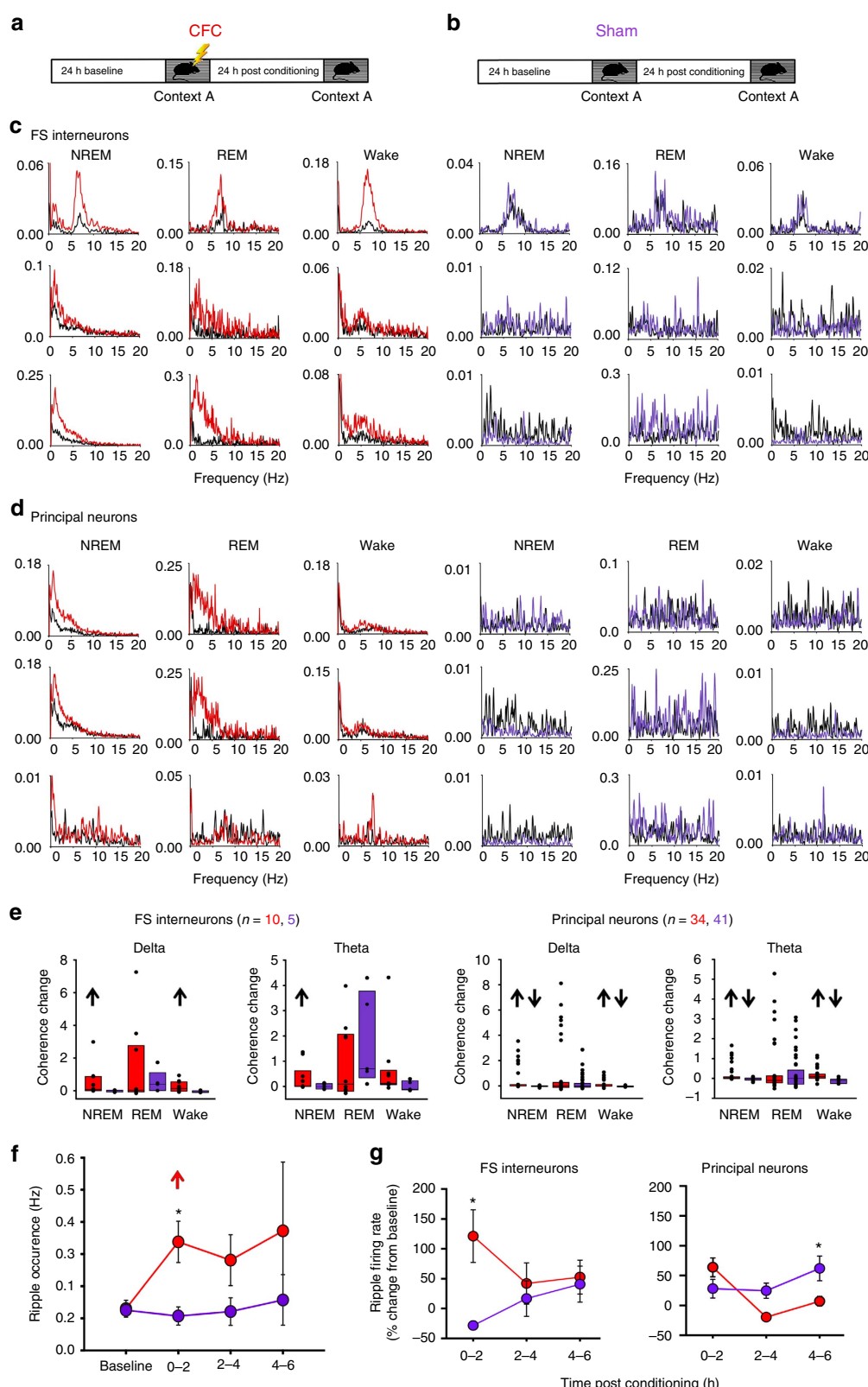

of mCherry-expressing controls (Fig. 3b). Taken together, these data show that CA1 PV+ interneurons play an essential role in CFM consolidation.

**PV+ interneuron-driven oscillations predict memory recall.** To determine how PV+ interneurons coordinate CA1 network activity during memory consolidation, we first characterized changes in CA1 LFP activity over the first 6 h following CFC. LFP power spectra during rapid eye movement (REM) sleep and non-REM (NREM) sleep are shown for a representative hM4Di-expressing mouse in PV+ interneuron inhibition (CNO) and control (vehicle) conditions (Fig. 4a,b; see also Supplementary Fig. 5a). Under control conditions, CA1 spectral power in delta (0.5–4 Hz) and theta (4–12 Hz) frequency bands increased dramatically during post-CFC REM and NREM (but not wake; Supplementary Fig. 5). These changes in spectral power were evident in data averaged across states, although there was some evidence that they waxed and waned across bouts of sleep (Fig. 4g). Delta and theta increases were transient (generally returning to baseline levels after ~6 h; Supplementary Fig. 5b), and were blocked by CNO administration (Fig. 4c,e, respectively; see also Supplementary Fig. 7). Critically, the degree to which NREM delta and theta activity increased in the first 6 h following CFC predicted individual animal's subsequent behavioural performance (that is, context-specific freezing measured 24 h post CFC; Fig. 4d,f, respectively).

Because PV+ interneurons are implicated in generation of CA1 SPWR oscillations, and NREM SPWRs are hypothesized to play a role in memory consolidation, we also assessed the effects of PV+ interneuron inhibition on SPWRs in the hours following CFC[26–28]. Under control conditions, both the amplitude and frequency of ripple (150–250 Hz) oscillations increased significantly over the first 2–4 h of post-CFC NREM sleep; these changes were also blocked by PV+ interneuron inhibition (Fig. 5). Together, these data demonstrate that PV+ interneurons normally coordinate multiple sleep-associated CA1 network oscillations in the hours following learning, and suggest that these oscillations in turn promote CFM.

**PV+ interneurons coordinate ensembles during consolidation.** Delta, theta and ripple oscillations could play a critical role in regulating spike timing between neurons, a process which is likely essential for memory consolidation[7,29,30]. To determine whether PV+ interneurons also play a role in coordinating CA1 neuronal ensemble activity in the context of CFM consolidation, we characterized neuronal functional connectivity patterns before and after CFC. To do this, we divided spike trains recorded from CA1 neurons into 1 min time bins, and compared spike train relationships among neurons within each time bin using a previously developed metric referred to as average minimum temporal distance (AMD)[31]. AMD characterizes functional connectivity relationships between pairs of neurons, based

on the temporal proximity of their respective spike trains (Fig. 6a; see also Supplementary Fig. 9). Using AMD, we generated pairwise functional connectivity matrices for all stably recorded CA1 neurons. These matrices were then compared between successive time intervals (Fig. 6b). Based upon mean minute-to-minute similarity of these matrices, we were able to quantify stability of the CA1 network across baseline and post-CFC recording periods. A representative baseline similarity trace is shown in Fig. 6c for a mouse expressing hM4Di-mCherry. Changes in CA1 network stability measured across 24 h after CFC (that is, from 24-h baseline) were compared between treatment groups. In vehicle-treated mice, CA1 network stability was significantly enhanced following CFC—a change which was present in both NREM sleep and wake. Increases in stability were blocked by PV+ interneuron inhibition (Fig. 6d). NREM stability following CFC was correlated with the occurrence of NREM ripples (particularly in the first 2 h following training; Fig. 6e), but was not predicted by NREM theta or delta changes (N.S., Spearman rank order, data not shown). Destabilization of functional connectivity patterns by post-CFC PV+ interneuron inhibition was not due to gross dysregulation of CA1 neuronal activity, as firing rates were not significantly altered by CNO treatment (Supplementary Fig. 8). Together this suggests that PV+ interneurons play a critical role in stabilizing communication patterns throughout the CA1 network in the hours following CFC.

**PV+ interneurons coordinate ensemble reactivation over time.** To visualize how CA1 network communication patterns change in the hours following CFC, we generated a functional similarity matrix (FSM; schematized in Fig. 7a) for each recording. The FSM illustrates the degree of similarity between a network's functional connectivity pattern at a given time point and at all other time points across a recording. FSMs are shown for NREM epochs a representative hM4Di-expressing mouse at baseline and for the first 6 h after CFC following either vehicle or CNO administration, respectively (Fig. 7b). Distributions of similarity values were quantified and compared across baseline and post-CFC recording periods (Fig. 7c). In vehicle-treated mice, there was more frequent and consistent repetition of specific network functional connectivity patterns throughout the first 6 h of post-CFC NREM, shown as a rightward shift in the similarity distribution after CFC (Fig. 7c,d; see also Supplementary Fig. 10). As was true for delta and theta LFP power changes, stable repetition of network patterns waxed and waned (and in some cases, appeared to do so in concert with LFP power changes; see Supplementary Fig. 11). Such changes in connectivity patterns across time were not seen when PV+ interneurons were inhibited. These findings show that in addition to promoting network oscillations after learning, PV+ interneurons promote consistent reactivation of CA1 neural ensembles over time.

**Figure 1 | Coherence changes in CA1 neurons during CFM consolidation.** (**a**,**b**) Experimental paradigm. Male C57BL/6J mice were recorded for a 24-h baseline period in their home cage, starting at lights-on, after which they either underwent single-trial CFC in a novel recording chamber (**a**) or sham training (**b**) (n = 5 mice per group). Afterward, all mice were returned to their home cage for an additional 24 h of recording. Context-specific freezing was assessed as a measure of CFM consolidation 24 h later. (**c**,**d**) Coherent firing is shown for three representative FS interneurons (**c**) and three representative principal neurons from the same site (**d**) recorded in CA1 over the first 6 h of baseline (black) or the first 6 h following either CFC or sham conditioning (red or purple, respectively). Coherent firing is shown separately for periods of NREM, REM and wake. (**e**) Quantification of post-CFC changes in delta and theta coherence (from baseline) for FS interneurons and principal neurons. Box plots indicate 25th, 50th and 75th percentile values. ↑↓ indicates relative increases or decreases of P < 0.05 for post CFC or post Sham versus baseline, Wilcoxon signed rank test. (**f**) Mean (±s.e.m.) frequency of SPWR events for CFC and Sham mice during the first 6 h of baseline recording, and following CFC or sham conditioning. ↑ indicates relative increase of P < 0.05 for post CFC versus baseline, *indicates P < 0.05 post CFC versus post Sham, Holm-Sidak post hoc test. (**g**) Mean (±s.e.m.) firing rate changes (from baseline values) during SPWR events. *indicates P < 0.05 post CFC versus post Sham, Holm–Sidak post hoc test.

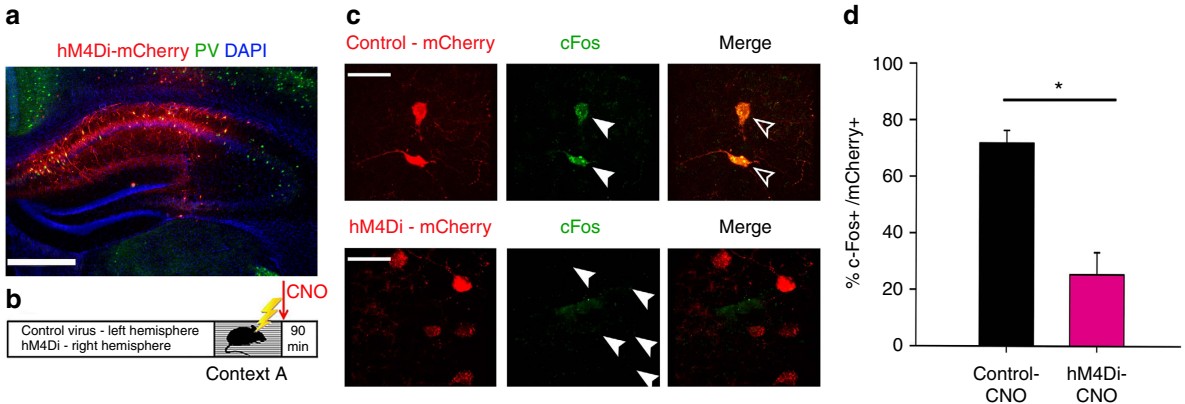

**Figure 2 | Pharmacogenetic inhibition of PV + FS interneurons.** (**a**) Representative image for expression of AAV-hM4Di-mCherry in CA1 PV + interneurons. Scale bar, 500 µm. (**b**) Experimental paradigm: *Pvalb-IRES-CRE* mice ($n = 5$) expressing hM4Di-mCherry in right-hemisphere CA1 and control (mCherry) virus in left-hemisphere CA1 underwent single-trial CFC were immediately administered CNO (0.3 mg kg$^{-1}$, *i.p.*) and were returned to their home cage for 90 min before killing. (**c**) CNO-mediated inhibition of PV + interneurons was confirmed by comparing the proportion PV + interneurons, which were c-Fos-immunopositive 90 min post-CFC between right (hM4Di-mCherry) and left (control) hemisphere. (left) mCherry expression in left and right hemispheres (top and bottom, respectively). Scale bar, 60 µm. (centre) c-Fos staining in cell bodies (green); locations of mCherry-expressing cell bodies identified by solid arrowheads. (right) Merged images showing c-Fos and mCherry expression (empty arrowheads indicate cells expressing both c-Fos and mCherry). (**d**) Fewer mCherry-labeled neurons expressed c-Fos in the hM4Di-expressing right hemisphere compared with the left (control) hemisphere (values indicated mean ± s.e.m./hemisphere). *indicates $P < 0.001$, Student's *t*-test, error bars are s.e.m.

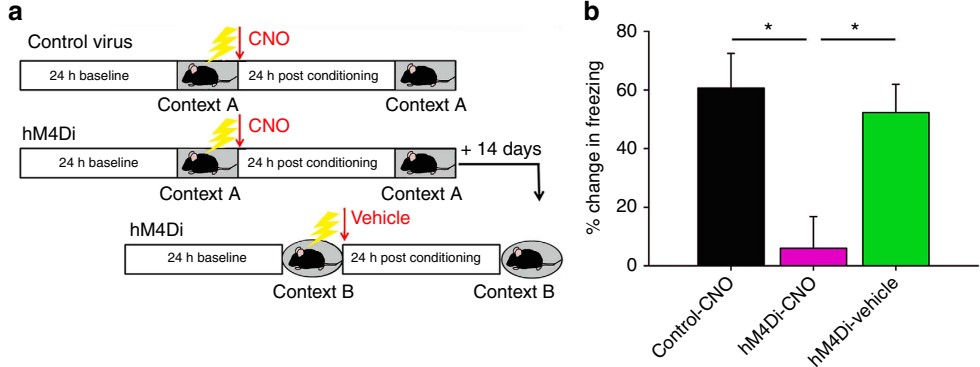

**Figure 3 | Pharmacogenetic inhibition of CA1 PV + interneurons blocks CFM consolidation.** (**a**) Experimental paradigm: mice expressed either hM4Di or mCherry ($n = 5$ for each group) bilaterally in CA1. Following a 24-h baseline recording period in their home cage, mice were placed into a novel recording chamber (Context A) for single-trial CFC. They were subsequently administered CNO and returned to their home cage. Context-specific freezing was assessed as a measure of CFM consolidation 24 h later. Two weeks later, hM4Di-expressing mice again underwent single-trial CFC—this time in a second, dissimilar context (Context B), after which they were administered vehicle (DMSO). (**b**) CFM in Context A was significantly reduced in mice with post-CFC inhibition of PV + interneurons. These same mice showed normal CFM in Context B when PV + interneurons were not inhibited. *indicates $p = 0.006$ one-way ANOVA, $P < 0.05$ Holm–Sidak *post hoc* test for hM4Di-CNO ($n = 6$) versus Control-CNO ($n = 4$) and versus hM4Di- vehicle ($n = 6$). Control-CNO versus hM4Di-vehicle *N.S.* All error bars are s.e.m. ANOVA, analysis of variance.

**PV + interneuron-driven rhythms promote network stability.** We next tested whether rhythmic PV + interneuron activity is sufficient to augment CA1 oscillations and stabilize CA1 ensembles. To do this, we recorded network dynamics among CA1 neurons before, during, and after rhythmic optogenetic stimulation of PV + interneurons in anaesthetized animals (Supplementary Fig. 12). In transgenic mice expressing Channelrhodopsin 2 (ChR2) in PV + interneurons (*PV:ChR2*), stimulation across a range of frequencies led to rhythmic activation of FS interneurons, and rhythmic inhibition of neighbouring principal neurons (Supplementary Fig. 13a). Stimulation also led to frequency-specific, rhythmic activity in the CA1 LFP and enhanced neuronal spike-field coherence at the stimulation frequency (Supplementary Fig. 13a; Supplementary Fig. 15a). In contrast, stimulation above (18 Hz) or below (2 Hz)

this range had relatively modest effects on spike-field coherence. Because PV-expressing interneurons outside CA1 can contribute to CA1 network oscillations[32], we repeated these experiments in AAV-transduced *Pvalb-IRES-CRE* transgenic mice with CA1-targeted expression of ChR2. In these mice, rhythmic PV + interneuron activation led to similar increases in CA1 rhythmic firing and LFP rhythmicity (Fig. 8a,b and Supplementary Fig. 14). This was true both under anaesthetized conditions (Fig. 8) and in awake, behaving mice (Supplementary Fig. 16). Again, the strongest effects on spike-field coherence were seen for stimulation frequencies between 4 and 10 Hz (Fig. 8c).

To test how optogenetically induced coherent firing affected stability of communication patterns between CA1 neurons, we quantified stability of functional connectivity between each neuron and all of its neighbours (see Methods for details) at

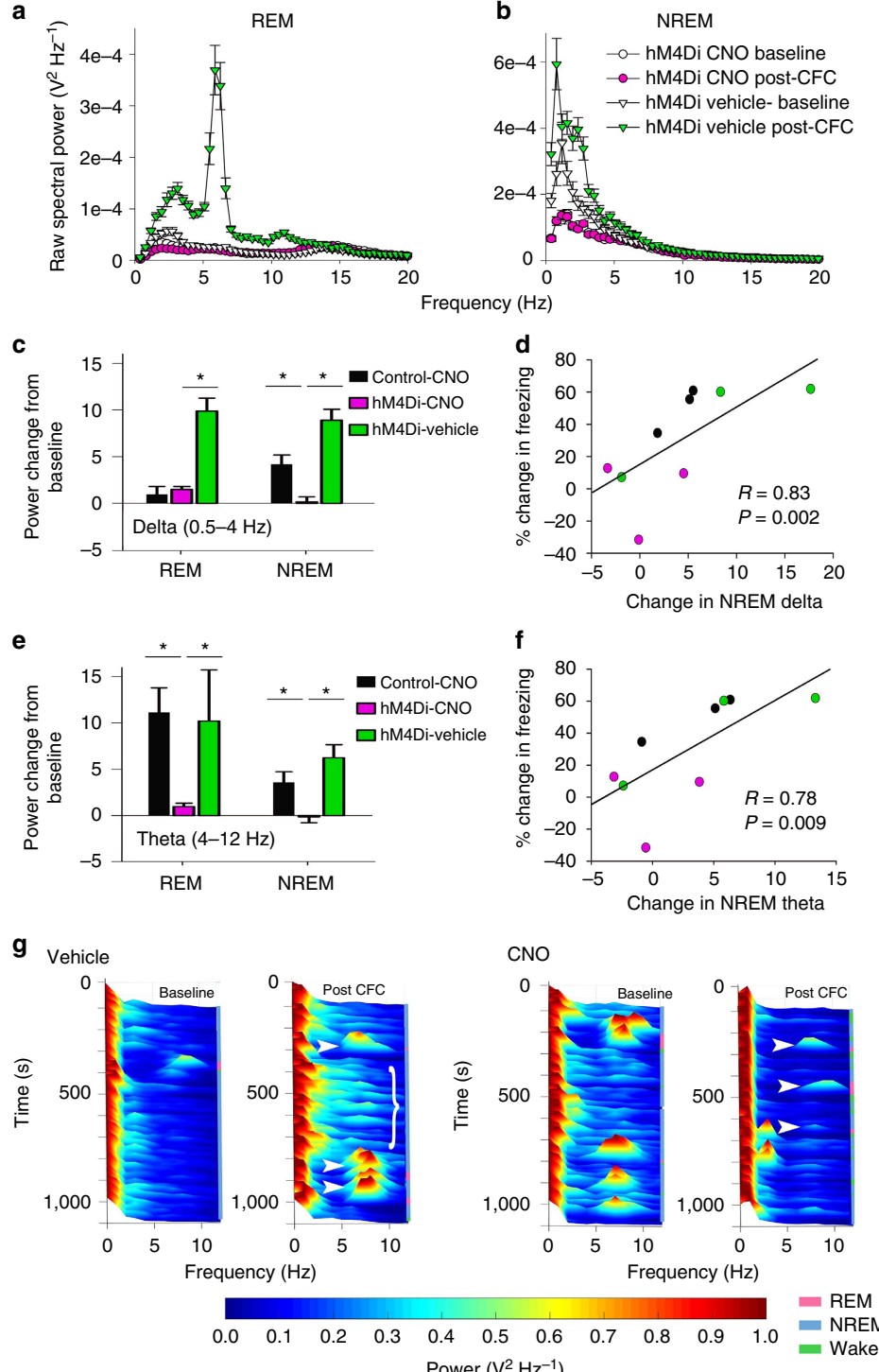

**Figure 4 | Inhibition of PV + interneurons disrupts the augmentation of CA1 delta and theta oscillations associated with successful CFM consolidation.** (**a,b**) Raw CA1 LFP spectral power in REM and NREM sleep from a representative hM4Di-expressing mouse over the first 6 h post-CFC, following administration of CNO (mean ± s.e.m. of 26 local field values from a representative animal; pink circles) versus vehicle (green triangles). (**c,e**) Post-CFC increases in delta and theta spectral power (from baseline) were present in the first 6 h of post-conditioning sleep, in the two control conditions; these increases were blocked by PV + interneuron inhibition. *indicates $P < 0.01$, Holm–Sidak *post hoc* test. Mean ± s.e.m. shown for LFP values for all animals across a given treatment (hM4-CNO, $n = 74$; hM4-vehicle, $n = 74$; Control-vehicle; $n = 58$). (**d,f**) Increases in NREM delta and theta power over the first 6 h post CFC were correlated with subsequent CFM consolidation (values indicate mean LFP changes per mouse; Spearman rank order). (**g**) Representative LFP spectrograms showing spectral power in delta and theta bands across 1,000 s of recording time (matched for time of day) during baseline and in the hours immediately following CFC. In vehicle-treated mice, following CFC there was greater power in delta and theta bands in NREM (bracket) and dramatically increased theta power during REM bouts (filled arrowheads). In CNO-treated mice, there was a suppression of theta in REM bouts (filled arrowheads) as well as a general suppression of 2–12 Hz activity across all states.

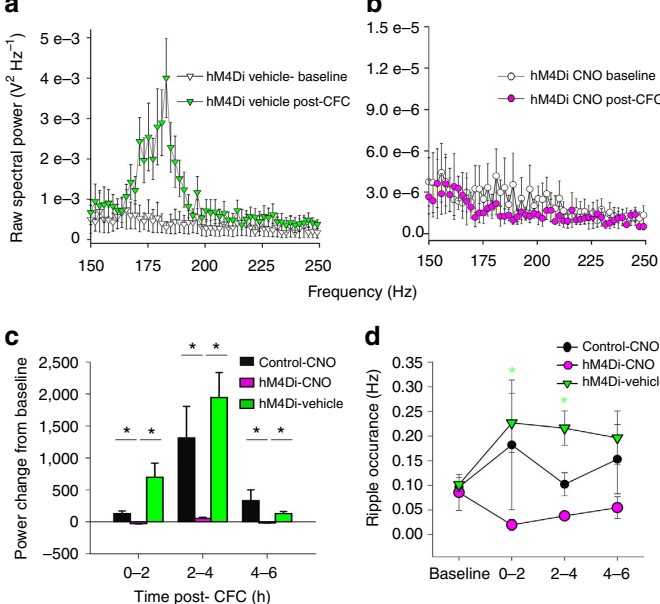

**Figure 5 | Inhibition of PV + interneurons disrupts augmentation of CA1 ripple oscillations during CFM consolidation.** (**a**,**b**) Ripple amplitude increased immediately following CFC + vehicle, but not CFC + CNO. Mean ± s.e.m. shown for LFP values for all animals across a given treatment (hM4-CNO, $n = 74$; hM4-vehicle, $n = 74$). (**c**) Post-CFC increases in ripple power from baseline were present in the first 4 h of post-conditioning sleep in the two control conditions; these increases were blocked by PV + interneuron inhibition. *indicates $P < 0.01$, Holm–Sidak post hoc test. (**d**) Post-CFC NREM ripple frequency increases were blocked by PV + interneuron inhibition. *indicates $P < 0.05$, Holm–Sidak post hoc test, vehicle versus CNO and versus baseline. All values indicate mean ± s.e.m.

baseline, during stimulation at various frequencies, and post stimulation. Stimulation of PV + interneurons across 2–10 Hz significantly increased network stability, relative to baseline (with the largest increases between 4 and 10 Hz; Supplementary Fig. 15d). Across the 4–10 Hz frequency range, the degree to which stimulation increased coherence predicted the change in stability of neurons' functional connectivity patterns (Fig. 8d and Supplementary Fig. 15e). Intriguingly, stability of network connectivity remained high relative to baseline even after the stimulation period ('post'; Supplementary Fig. 15d). This suggests that rhythmic synchronization of firing in CA1 has an effect on the network which can outlast the period of PV + interneuron stimulation.

To test the longer-term effects of PV + interneuron-induced network coherence, we carried out two additional sets of experiments (in anaesthetized and awake mice, respectively) where stimulation at 7 Hz (that is, the middle of the 4–10 Hz range) was followed by least 2 h of post-stimulation recording (Supplementary Fig. 12b,c). This stimulation frequency evoked a robust increase in spike-field coherence in PV:ChR2 mice, but not in control mice expressing GFP in PV + interneurons (PV:GFP; Fig. 8f; Supplementary Fig. 13b; and Supplementary Fig. 15f). Stimulation of PV + interneurons at 7 Hz induced a significant increase in CA1 network stability that lasted through the entire post-stimulation period (Fig. 8f; Supplementary Fig. 15g; see also Supplementary Fig. 16c). This manipulation also induced a long-lasting increase in the strength of network connections (quantified as the average mean temporal distance between neuronal pairs' spike trains; see 'Methods' for more details; Fig. 8g; and Supplementary Fig. 15h). Not only could a long-term

change in network connection strength be induced in anaesthetized animals; it was also seen in awake, behaving animals for up to 2 h after cessation of 7-Hz stimulation (Supplementary Fig. 16d). Taken together, these data suggest that rhythmic coordination of CA1 network activity by PV + interneurons (at frequencies corresponding to those where increased network rhythmicity is seen following learning) has long-lasting effects on both the stability and strength of network connections.

## Discussion

Our current findings demonstrate the essential mechanistic role that PV + interneurons play in hippocampally mediated memory consolidation. We find that CA1 FS interneurons show natural changes in their firing dynamics during active memory consolidation, in the hours following single-trial CFC. We also find that in the absence of CA1 PV + interneuron activity, (1) CFM consolidation is completely blocked, (2) normal post-CFC enhancements in delta, theta and ripple oscillations are lost and (3) normal post-CFC stabilization of network communication patters is lost. Rhythmic activity in PV + interneurons is sufficient to drive coherent firing in neighbouring CA1 neurons and to stabilize firing relationships among them, leading to long-lasting changes in network communication patterns and the strength of network connections. Thus the role of CA1 PV + interneurons in CFM consolidation is not only essential, it may also be instructive—promoting the changes to neuronal ensemble-level dynamics seen in the hours following learning.

While relatively few in number (~2.6% of the CA1 neuronal population), PV + interneurons each innervate large numbers of pyramidal cells[19]. Thus, from an anatomical standpoint, they are uniquely well-suited to coordinate ensemble activity and spike timing within neuronal ensembles. PV + interneurons were recently shown to play a critical role in driving theta and ripple oscillations in ex vivo hippocampal slices[20,33]. We have previously shown that the amplitude of CA1 theta and delta oscillations naturally increases following single-trial CFC in C57BL/6J mice. However, whether (and how) these network oscillations promote memory consolidation has remained a matter of speculation[23]. Here we show that (1) PV + interneurons drive enhancements in theta, delta, and ripple oscillations in the hours following learning and (2) these oscillations can have long-lasting effects on the strength of network connections and the stability of CA1 ensemble activity.

Reactivation and sequential replay have been widely reported in CA1 after spatial learning, yet little is known about what drives these processes, or what their functional significance might be[7,10,12,13]. Here we employ a new set of metrics, which can be used to detect network changes after single-trial learning events such as CFC, and do not rely on specific features such as sequential place cell activation. For these reasons, they may be more broadly applicable for studies of memory consolidation when compared with other metrics. By quantifying changes in functional connectivity between CA1 neurons in the hours following single-trial CFC (when de novo memory consolidation is clearly occurring), our data add to what is known about reactivation and replay in several ways. First, we show that consolidation-related changes in CA1 network dynamics can be measured over a longer time window than previously reported for either sequential replay or ensemble reactivation (that is, over several hours following learning, versus tens of minutes)[34]. This timescale more closely matches what is known about CFM consolidation, in which memories remain susceptible to disruption for several hours following learning[6,24,25,35]. Second, we find that stabilization of functional connectivity patterns is associated with reliable reactivation of

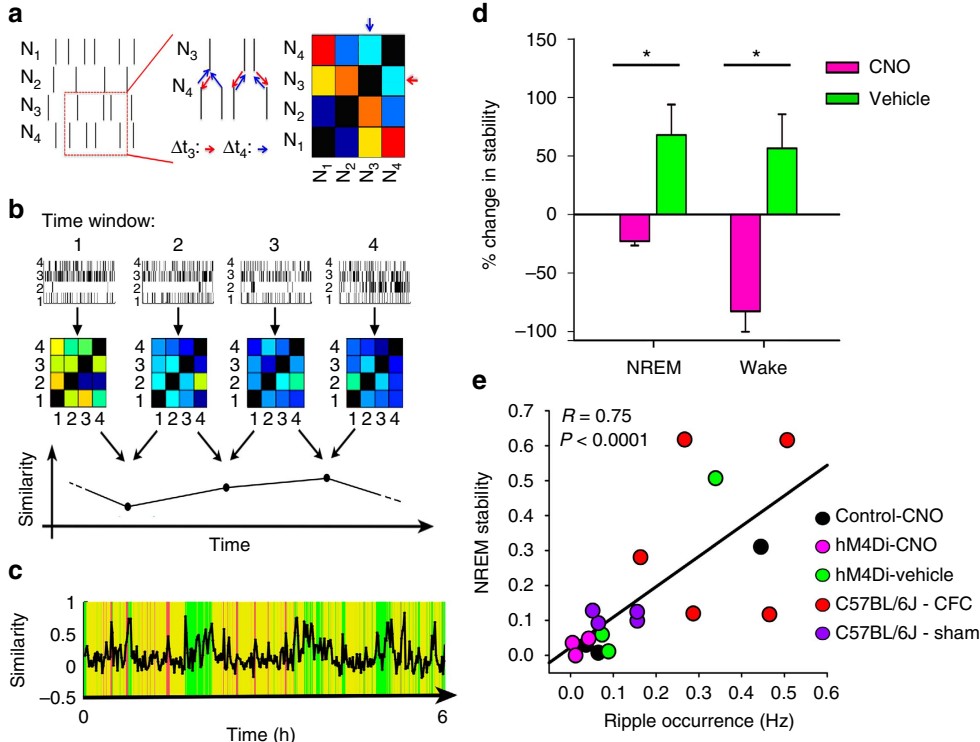

**Figure 6 | Inhibition of PV + interneurons disrupts learning-induced stabilization of CA1 network dynamics.** (**a**) AMD is calculated between spikes of two neurons. An example of AMD calculations for neurons '3' and '4'. The closest spikes of neuron '4' to each given spike of neuron '3' (red arrows) and vice versa (blue arrows). Pairwise AMD values were calculated for every 1-min interval, based on spike trains of stably recorded neurons. From these values, a functional connectivity matrix is generated, which represents the pattern of functional connectivity at any time point. Pairs with smaller AMD values are more likely to be functionally connected (shown in red in the matrix) than those with larger AMD values (shown in blue in the matrix). (**b**) Functional connectivity matrices were constructed for every 1-min interval of recording, based on spike rasters from all stably recorded neurons ($n = 5$–16 neurons/mouse, three mice/condition). A comparison of matrices at adjacent time points yields a similarity value, which is plotted across the entirety of a recording period. (**c**) Similarity values are shown over the first 6 h of baseline recording in a representative animal. Yellow indicates NREM sleep, green: wake, pink: REM sleep. (**d**) In the 24 h following CFC, mean ( ± s.e.m. shown) minute-to-minute CA1 network stability increased during both NREM and wake (REM stability could not be calculated due to the low number of REM epochs ≥2 min in duration). CNO inhibition of PV + interneurons decreased stability across this same time period. *indicates $P < 0.05$, Wilcoxen rank-sum test. (**e**) Across individual mice, post-CFC (or post-Sham) NREM stability was predicted by NREM ripple frequency in the first 2 h following training (Spearman rank order).

CA1 ensemble activity patterns in the hours after learning. Third, we show that disrupting PV + interneurons' activity disrupts stabilization of the CA1 network and the consistent reactivation of CA1 neural ensembles over time. Finally, we show that the rhythmic activation of PV + interneurons (which is naturally augmented following learning) is sufficient to stabilize CA1 network dynamics over behaviourally relevant timescales (that is, hours).

Our data suggest that rhythmic activity in PV + interneurons is capable of promoting long-term 'functional' plasticity in the CA1 network. Whether, and how, PV + interneuron-driven dynamics affect 'structural' plasticity in CA1 remains an open question. Recent studies suggest that these neurons themselves undergo synaptic remodelling as a function of memory formation[36,37]. Intriguingly, disruption of sleep in the hours following learning has also been shown to impair structural plasticity in CA1 pyramidal neurons' dendritic spines[38]. One possibility is that the sleep-associated coordination of activity among CA1 neurons, over a timescale of hours, drives such structural changes by regularizing spike-timing relationships (that is, via spike timing-dependent plasticity).

In summary, our present data demonstrate that PV + interneuron-mediated network oscillations are associated with successful memory consolidation. We show for the first time that these oscillations can: (1) drive reliable ensemble reactivation;

(2) stabilize patterns of neuronal communication; and (3) induce long-lasting changes in the strength of functional connectivity relationships between CA1 neurons. We conclude that these neurons may provide instructive offline reinforcement of newly learned information by coordinating neuronal activity patterns across the CA1 network.

## Methods

**Mouse handling and surgical procedures.** All animal husbandry and surgical/experimental procedures were approved by the University of Michigan IACUC. With the exception of conditioning and fear memory testing, mice were individually housed in standard caging with beneficial environmental enrichment (nesting material and novel foods) throughout all experimental procedures. Lights were maintained on a 12 h:12 h light: dark cycle (lights on at 8:00), and food and water were provided *ad lib*. C57BL/6J mice (Jackson) were implanted with custom-built, driveable head stages with two bundles of stereotrodes for single unit/LFP recording and EMG wires for nuchal muscle electromyographic recording, as previously described[23]. The two stereotrode bundles were spaced ∼1.0 mm apart within right-hemisphere CA1 (relative to Bregma: 1.75–2.75 mm posterior; 1.5–2.5 mm lateral; and 1.0 mm ventral). CA1 recording sites for these experiments are shown in Supplementary Fig. 2.

At age 2–3 months, male *Pvalb-IRES-CRE* mice (B6;129P2-Pvalb$^{tm1(cre)Arbr}$/J; Jackson) received bilateral CA1 injections of AAV to express either the inhibitory receptor hM4Di (rAAV2/Ef1A-DIO-hM4Di-mCherry; UNC Vector Core: Lot # AV4,708) (ref. 21), or an mCherry reporter without hM4Di (control; rAAV2/Ef1A-DIO-mCherry; UNC Vector Core: Lot # AV4375FA) in a CRE-dependent manner. For both vectors, a volume of 1 µl was injected bilaterally via a 33-gauge beveled

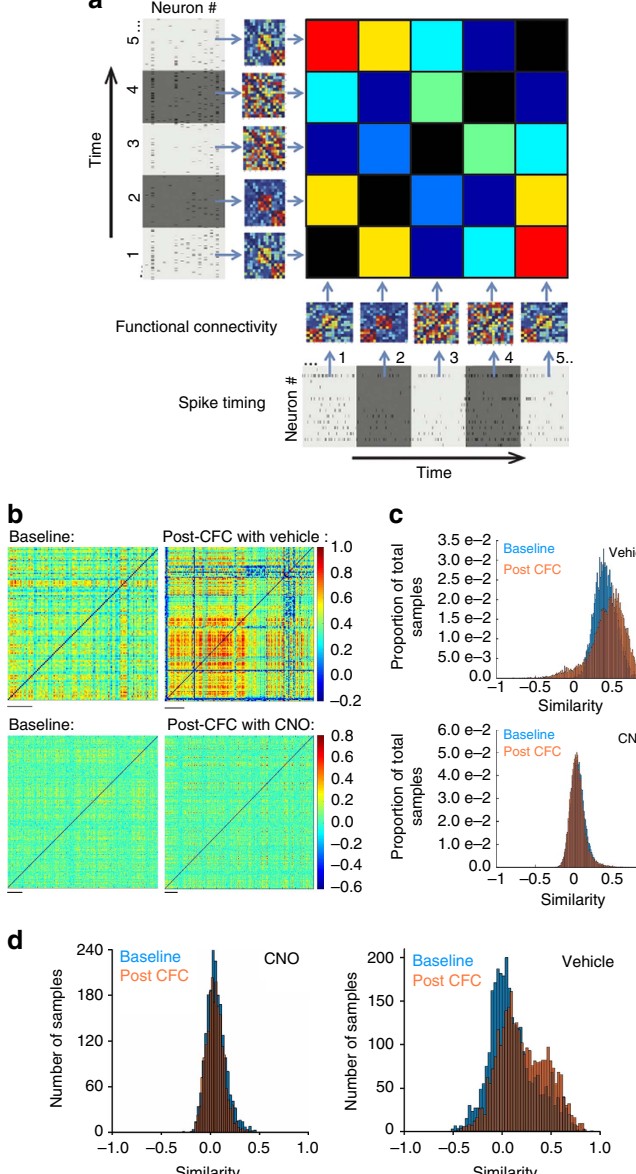

**Figure 7 | PV+ interneurons promote consistent reactivation of CA1 neural ensembles following learning.** (**a**) Generation of a FSM. The FSM displays the similarity of functional connectivity patterns (that is, matrices from Fig. 6b) across all time intervals. (**b**) NREM CA1 network FSMs from a representative hM4Di-expressing mouse at baseline, and over the first 6 h post CFC (vehicle and CNO conditions). Colour in the body denotes the degree of similarity between NREM functional connectivity patterns at any given time point in the recording, and NREM patterns at all other time points. Scale bars, 20 min of recording time. (**c**) Distributions of minute-to-minute similarity values (at baseline, and following CFC) for the data shown in **b**. (**d**) Distributions of NREM sleep similarity values from all stably recorded mice at baseline, and following CFC (n = 3/condition). For both (**c**,**d**), similarity distributions were significantly shifted to higher values following CFC in vehicle-treated mice, but not CNO-treated mice (P < 0.001, Mann–Whitney rank sum test).

syringe needle at a rate of 0.2 μl min⁻¹ (from Bregma 2.0 mm posterior, 2.0 mm lateral). They were subsequently implanted with driveable headstages as described above. CA1 recording sites for these experiments are shown in Supplementary Fig. 4.

**Recording procedures.** Mice were prepared for chronic stereotrode recording as described previously[23], either 1 week after implantation surgery (C57BL/6J) or 4 weeks after AAV transduction (*Pvalb-IRES-CRE*). Mice in their home cage were placed into sound-attenuated chambers (Med Associates) for recording of CA1 activity during natural sleep/wake states and CFC (see below). Stereotrode bundles were slowly advanced into CA1 in 10–20 μm steps over a period of 3–7 days, until stable recordings were obtained. During this period, mice were habituated to daily handling and the recording apparatus (including tethering via lightweight recording cables (Plexon, Inc.)). All experiments began with a 24-h baseline recording period, starting at lights-on, and continued throughout subsequent, single-trial CFC, a 24-h post-training CFM consolidation period, and subsequent CFM testing in the conditioning chamber. Neuronal spike and LFP signals were acquired by differentially filtering data from each electrode wire (bandpass 300 Hz–8 KHz and 0.5–300 Hz, respectively); these data were digitized and amplified using Plexon Omniplex hardware and software as described previously[23].

**Conditioning and pharmacogenetic inhibition.** Following 24 h baseline recording (beginning within 1 h of lights-on), mice underwent single-trial CFC or sham conditioning. Mice were placed in a novel conditioning chamber with walls made of clear Plexiglas and a shock grid floor (Med Associates). CFC and sham-conditioned C57BL/6J mice (n = 5 mice per group) were placed in the same environmental context for training (Context A, below). CFC mice were allowed to explore the conditioning chamber for 2.5 min before receiving a 2 s, 0.75 mA foot shock via the grid floor. After the foot shock, mice remained in the conditioning chamber for an additional 28 s. Sham mice received no foot shock during 3 min of exploration. *Pvalb-IRES-CRE* mice underwent CFC in either Context A or Context B as described. Each mouse was then immediately given a 0.04 ml i.p. injection of either 0.3 mg kg⁻¹ clozapine-*N*-oxide (CNO; Santa Cruz Biotechnology; n = 6 mice) dissolved in DMSO, or an injection of DMSO alone (vehicle; n = 5 mice).

Following CFC or sham conditioning, all mice were then left undisturbed in their home cage for the next 24 h, after which they were returned to the conditioning chamber for 5 min to assess CFM. Contextual freezing was quantified as a change in the per cent of time spent in freezing behaviour from pre-shock baseline during CFC or during sham conditioning. *Post hoc* scoring of videos for freezing behaviour was conducted by an observer blind to both the animals' viral expression and drug treatment. Behaviour was scored as freezing in the following conditions: crouched posture with an absence of all body movement save respiration (including an absence of head and whisker movement)[39,40].

For Context A, the chamber was cylindrical in shape and had vertical stripes over one half of the vertical surface. For Context B, the chamber was square in shape and had horizontal grating pattern on three out of the four walls. Chamber walls and floor were cleaned thoroughly with 70% ethanol both before and immediately following conditioning.

**Immunohistochemistry.** At the end of each experimental recording, mice were anaesthetized with isoflurane and all electrode sites were lesioned (2 mA, 3 s per wire) before perfusion and euthanasia. To verify CA1 electrode placement and confirm proper AAV-mediated expression relative to recording electrodes, the brain was post fixed and sectioned at 50 μm. To confirm AAV-mediated transgene expression to PV+ interneurons (for example, Fig. 2a), brain sections containing dorsal CA1 were immunostained with goat anti-parvalbumin (1:1,000; Abcam; ab11,427). Brain slices were mounted using DAPI Fluoromount-G (Southern Biotech).

To confirm CNO-mediated inhibition of PV+ interneurons, *Pvalb-IRES-CRE* mice (n = 5) expressing rAAV2/Ef1A-DIO-hM4Di-mCherry in right CA1 and rAAV2/Ef1A-DIO-mCherry in left CA1 underwent single-trial CFC at lights on. Immediately following CFC, each mouse was administered CNO, after which they were allowed *ad lib* sleep in their home cage. Ninety minutes later, each mouse was deeply anaesthetized with isofluorane, perfused with ice cold 0.1 M PBS, and perfused with 4% paraformaldehyde in PBS (PFA-PBS). 50 μm coronal brain sections were obtained for histochemistry using antibodies for c-Fos (1:6,250; Millipore; PC05). Images were collected for all brain slices containing virally transduced hippocampal regions. Quantification of c-Fos immunoreactivity in virally transduced PV+ interneurons was conducted by a scorer blind to treatment conditions, following previously published methods[41]. Briefly, for each brain slice, the number of CA1 neurons expressing mCherry (that is, those virally transduced neurons with CRE-mediated transgene expression) were first counted. CNO-induced changes in PV+ interneuron activity were assessed by quantifying the percentage of mCherry-expressing neurons which were also c-Fos immunopositive (# of cells c-Fos positive/ # of mCherry-positive cells). Comparisons were made between right (hM4Di) and left (control) hemispheres in the same mouse. Neurons with cell bodies that were out of the plane of focus were not included in quantification.

**Single-neuron discrimination and firing analysis.** Single-neuron data were discriminated offline using standard principle-component-based procedures (Offline Sorter; Plexon). Individual neurons were discriminated on the basis of spike waveform, relative spike amplitude on the two stereotrode recording wires, relative positioning of spike waveform clusters in three-dimensional principal component space, and neuronal subclass (for example, FS versus principal; Supplementary Fig. 1a,b). Single-neuron isolation was verified using standard techniques, that is, elimination of clusters with interspike interval (ISI) -based

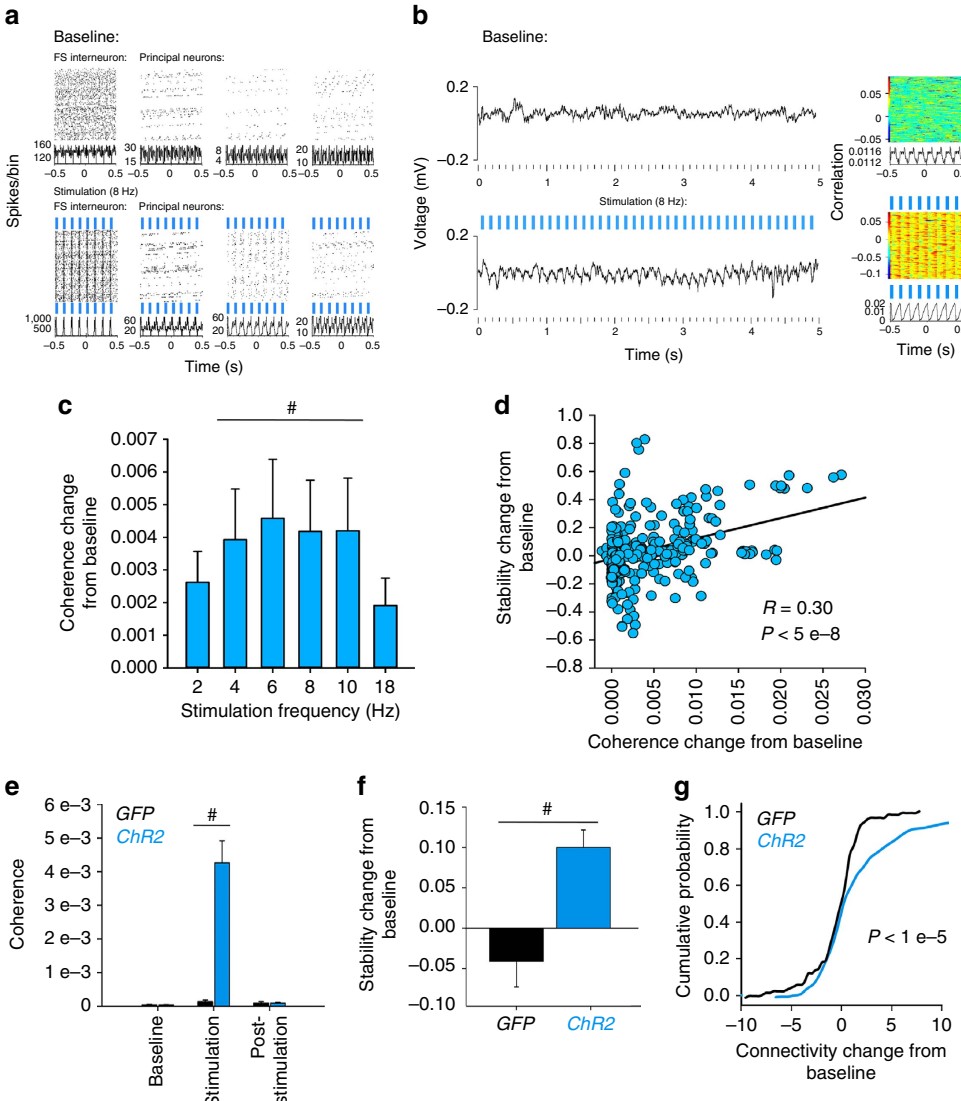

**Figure 8 | Coherent firing induced by optogenetic stimulation of PV+ interneurons increases CA1 network stability and connection strength.**
(**a**) Perievent firing rasters (top) and perievent firing histograms (bottom) for a representative CA1 FS interneuron and three neighbouring principal neurons recorded from a *Pvalb-IRES-CRE* mouse expressing ChR2. Firing is shown over 250 s of recording before and during rhythmic (8 Hz) 473 nm light stimulation of PV + interneurons. (**b**) A representative 5 s LFP trace (left) and perievent LFP raster (right) for one of the recording sites from **a** in baseline and stimulation conditions. (**c**) Changes in spike-field coherence (from baseline) induced by various frequencies of rhythmic PV + interneuron stimulation in virally transduced neurons. Significant increases (from baseline) were present at stimulation frequencies between 4 and 10 Hz. #indicates $P < 0.05$, Wilcoxon signed rank test. (**d**) Across 4–10 Hz stimulation frequencies, changes in spike-field coherence predicted changes in stability of functional connectivity for individual neurons (Spearmann rank order, $n = 320$ neurons). (**e**) Comparison of CA1 spike-field coherence across a 30-min baseline period, 30 min of 7 Hz stimulation, and 2 h or post-stimulation recovery, in mice with CA1 transduction of ChR2 (blue) or GFP (black). (**f**) Over the 2 h following 7 Hz stimulation, CA1 neuronal functional connectivity in mice transduced with ChR2 showed an increase in stability relative to baseline. For (**c,d**), #indicates $P < 0.05$, Wilcoxon signed rank test. For (**c,e,f**), all values indicate mean ± s.e.m. (**g**) Neuronal functional connectivity strength also showed an increase in ChR2-transduced mice following 7 Hz stimulation. A similar change that was not seen in GFP-transduced mice. $P$ value indicates results of Kolmogorov–Smirnov test.

absolute refractory period violations[42]. For assessing spike cluster quality, spike clusters on channels with multiple stably recorded single units were compared with one another, and with identified noise, in principal component space (Supplementary Fig. 1c). Waveform cluster separation was first validated using multivariate analysis of variance on the first three principal components ($P < 0.05$ for all sorted clusters), and further characterized using the Davies-Bouldin (DB) validity index (a metric with inter-cluster distance as the denominator, thus lower values indicating better cluster separation)[43]. The mean ( ± s.e.m.) DB value for all sorted waveform clusters (across all groups) was 0.36 ± 0.02 (for spike clusters versus spike clusters) and 0.46 ± 0.02 (for spike clusters versus noise), which compares favourably with DB values from single-unit data used in other studies[44,45] (distributions of discrimination values shown in Supplementary Fig. 1c). Only those neurons that (1) met the criteria described above and (2) were reliably discriminated and continuously recorded throughout each experiment

(that is, those stably recorded across both 24-h baseline and 24-h post CFC recording) were included in firing rate analyses from behaving mice ($n = 10$ FS and 34 principal neurons for CFC C57BL/6J mice, $n = 5$ FS and 41 principal neurons for sham C57BL/6J mice; $n = 27$ neurons for Control + CNO, 35 neurons for hM4Di + CNO, and 21 neurons for hM4Di + vehicle *Pvalb-IRES-CRE* mice). Learning-induced coherence changes were calculated by directly comparing state-specific firing coherence from 0.5–20 Hz across the first 6 h of baseline recording and the first 6 h following CFC or sham conditioning. Post-CFC firing rate changes were calculated over 6-h windows for each neuron as a per cent change from baseline, as described previously[23].

**Sleep/wake behaviour and LFP analyses.** Intra-hippocampal LFP and nuchal electromyography signals were used to categorize each 10 s interval of recording as

either REM, NREM or wake (Supplementary Fig. 3a) using custom software. REM was identified based on highly regular theta-frequency activity in the LFP and low EMG activity, NREM was characterized by relatively high-amplitude, low-frequency LFP oscillations, and wake was characterized by high-frequency, low-amplitude LFP activity and relatively high EMG activity. The proportion of time spent in REM, NREM and wake (and mean bout duration for each state) was calculated during the baseline and post-CFC recording periods for each mouse using standard conventions.

Raw LFP power (0–300 Hz, in 0.4 Hz bands) was calculated on each channel, where stable neuronal spike data was obtained. Spectral power was quantified from raw LFP traces within each 6-h time window at baseline and post-CFC. Changes in power were calculated within each frequency bin as ((power post CFC − power at baseline)/power at baseline). These values were summed across frequency bands for recording site at delta (0.5–4 Hz), theta (4–12 Hz) and gamma (25–50 Hz) for subsequent analysis. Ripples were quantified within NREM sleep in 2-h windows, using previously described methods[28]. Briefly, LFPs were bandpass filtered (150–300 Hz) and ripple events were automatically detected using a threshold of six or more consecutive cycles of an oscillation with a voltage ± 2 s.d. from baseline signal mean. To compare ripple occurrence between baseline and post-CFC recording periods, ripples were detected across recording periods using the same (baseline) voltage threshold. Spectral power was calculated within detected ripple intervals at baseline and post CFC; changes in power were calculated as described above for comparisons of ripple amplitude between groups.

Spectrograms of 0–12 Hz PSD activity were generated in MATLAB. PSD data for representative LFPs were calculated in 1-s windows, and PSDs were convolved over 30 s at 1 Hz. PSDs were normalized over the entirety of the selected temporal window and expressed as $V^2 Hz^{-1}$.

**Functional connectivity analysis.** Functional connectivity in CA1 was calculated using spike trains $\{S_1, S_2, …, S_n\}$ for $n$ stably recorded CA1 neurons. The functional relationship $FC_{ij}$ was calculated for each pair of neurons (for example, $i$-th and $j$-th neurons) by first calculating the average temporal proximity of spike trains $S_i$ and $S_j$. The average distance from spike train $S_i$ to $S_j$ is given by the AMD[31]. $AMD_{ij} = \frac{1}{N_i} \sum_k \Delta t_k^i$, here $N_i$ is the number of events in $S_i$ and is the temporal distance between event $k$ in $S_i$ to the nearest event in $S_j$. This value was compared with the expected sampling distance of train $S_j$, $\mu_j$ and the difference between the two was then normalized to the expected variation in sampling distance of train $S_j$, $\sigma_j$. $\mu_j$ and $\sigma_j$, were calculated as follows, by integrating over the sampling minimum distance distribution of $S_j$. The expected mean sampling distance and s.d. were calculated from ISIs in $S_j$ of length $L$. These values were quantified as the first two moments of minimal distance given by $\mu^L = <MD^L> = (1/4)L$ and $<(MD^L)^2> = (1/12)L^2$. The probability of sampling an interval of length $L$ for a spike train of length $T$ is $(L/T)$. Thus we combined the intervals to give $\mu_j = <MD_j> = \sum_{\{L\}} p_L \mu^L = \frac{1}{T} \sum_{\{L\}} \frac{L^2}{4}$, and $\langle (MD_j)^2 \rangle = \frac{1}{T} \sum_{\{L\}} \frac{L^3}{12}$. The expected s.d. is given by $\sigma_j^2 = <(MD_j)^2> - <MD_j>^2$. Thus the functional connectivity between neurons $i$ and $j$ was calculated as $FC_{ij} = \sqrt{N_i} \frac{\mu_j - AMD_{ij}}{\sigma_j}$. This value represents the significance of mean temporal proximity of $S_i$ to $S_j$ after taking into account the spiking distributions of $S_j$.

Performance of AMD is compared with cross-correlation-based methodology in Supplementary Fig. 9. The two methods were first used on two simulated spike trains. The first 30-spike spike train was generated randomly over a 1 s time interval. The second train was created by introducing a variable temporal jitter (1–100 ms) to the first spike train. Cross-correlation was based on convolution of the two trains, using Gaussians of variable width (2.36 ms ≤half-width ≤35.4 ms). Cross-correlation significance was calculated at 0 time delay, using bootstrapping based on 100 randomized (shuffled ISI) spike trains, and s.d. of significance was estimated over 100 repetitions of the randomized spike train. For comparison, AMD significance was calculated both analytically (as described above) and using the same bootstrapping technique.

**Functional similarity and stability analysis.** The functional connectivity of the entire recorded CA1 population for a given mouse was calculated over successive 1 min time segments (this is schematized as the functional connectivity matrix in Fig. 6). Adjacent time segments' connectivity patterns were then compared using cosine similarity, $C_{AB} = \cos \theta_{AB} = \frac{<A,B>}{\sqrt{<A,A>*<B,B>}}$, as a measure of the overlap between the two values. A cosine similarity value of 1 denotes no change in the network and a value of 0 indicates that the two networks are completely unrelated. Cosine similarity was calculated across the entire series of adjacent time windows, over the entire 24 h of baseline and the entire 24 h of post CFC recording. Average cosine similarity values (for adjacent time windows in a given behavioural state, for example, NREM or wake) constituted the functional stability metric for that state. Due to the relative infrequency and short duration of REM epochs (which typically lasted less than 1 min each) there were an insufficient number of successive recording epochs to reliably calculate network stability changes specifically within REM. Post-CFC changes in CA1 network stability were calculated separately for NREM and wake as (mean stability baseline—mean stability post CFC)/mean stability baseline × 100. FSMs (Fig. 7) were calculated using the algorithm described above; however,

**Optogenetic stimulation of PV+ interneurons.** Two groups of mice were used to assess the effects of rhythmic stimulation of PV+ interneurons. For experiments shown in Supplementary Figs 13 and 15, Pvalb-IRES-CRE mice were crossed to either B6;129S-Gt(ROSA)26Sor[tm32(CAG-COP4*H134R/EYFP)Hze]/J or B6.Cg-Gt(ROSA)26Sor[tm6(CAG-ZsGreen1)Hze]/J transgenic mice (Jackson) to express ChR2 (PV:ChR2) or eGFP (PV:GFP), respectively, in a CRE-dependent manner in PV+ interneurons. At age 2–5 months, male PV:ChR2 ($n = 7$) or PV:GFP ($n = 6$) mice were anaesthesized with isofluorane (0.5–0.8%) and 1 mg kg$^{-1}$ chlorprothixene (Sigma). Mice were head-fixed and a 1 mm × 1 mm matrix multi-electrode (250 μm electrode spacing; Frederick Haer Co. (FHC), Bowdoin, ME) was slowly advanced into CA1 until stable recordings (with consistent spike waveforms continuously present for at least 30 min before baseline recording) were obtained. An optical fibre was placed adjacent to the recording array for delivery of 473 nm laser light (CrystaLaser). Power output at the fibre tip was estimated at 3–10 mW for all experiments.

For the first set of recordings (Supplementary Fig. 12a), CA1 neurons were recorded over a 15-min baseline period, after which PV+ interneurons were stimulated over multiple successive 15-min periods with a range of frequencies (2–18 Hz, 40 ms pulses). The various stimulation frequencies were presented in a random interleaved manner, during which neuronal activity continued to be recorded. Following the final stimulus period, neurons were recorded for an additional 15 min to measure changes in network activity from baseline. Only those neurons recorded throughout the entire experiment (baseline + optogenetic stimulation + post stimulation) were included in analyses of optogenetically induced spike-field coherence and network stability changes.

For the second set of recordings (Supplementary Fig. 12b), neurons were recorded for a longer (30 min) baseline period, a 30 min stimulation period (7 Hz; 40 ms light pulses), and a longer post-stimulus interval lasting 2 h. Only those neurons recorded throughout the entire experiment (baseline + optogenetic stimulation + post stimulation) were included in analyses of optogenetically induced spike-field coherence and network stability changes.

To ensure that effects of optogenetic manipulation were not due to expression of ChR2 outside of CA1 in PV:ChR2 mice, male Pvalb-IRES-CRE mice (aged 2–3 months, $n = 7$) received bilateral CA1 injections of AAV2/9.EF1a.DIO.hChR2(H134R)-EYFP.WPRE.hGH (Penn Vector Core). Four to six weeks after AAV transduction, these mice underwent the same optogenetic manipulations described above; data from these experiments are shown in Fig. 8 and Supplementary Fig. 14.

To test whether rhythmic stimulation of PV+ interneurons could induce CA1 rhythms and long-term changes in the CA1 network in the absence of anaesthesia, two additional groups of AAV-transduced mice (ChR2- and GFP-expressing, $n = 3$/group, aged 2–3 months) were implanted with stereotrode recording arrays as described above, with optical fibres targeting CA1. Once stable neuronal recordings were achieved, these mice were recorded for two h under awake, behaving baseline conditions, across a 2-h period of 7 Hz stimulation, and over a 2-h post-stimulus interval (Supplementary Fig. 12c). Data from these experiments are shown in Supplementary Fig. 16.

Following all optogenetic experiments, mice were perfused and brains were processed for histological assessment as described above. AAV-mediated gene expression in CA1, optic fibre position and electrode position were validated before the data analysis.

**Data analysis for optogenetic recordings.** Single-neuron discrimination and sorting was conducted as described above for stereotrode recordings, except that relative spike amplitude across channels was not used as a criterion for discrimination. For all spike-field coherence analyses, LFP traces for each neuron's corresponding electrode recording site were filtered across a narrow band (stimulation frequency ± 0.2 Hz). Segments (500 ms) of this filtered LFP trace (centred around each spike) were averaged, and the amplitude (maximum–minimum voltage) of each mean trace was calculated as a spike-field coherence metric.

To calculate functional network stability for each recorded neuron (Fig. 8d; and Supplementary Fig. 15d,e), we first generated a functional connectivity matrix of all stably recorded CA1 neurons across successive 1-min time bins. Each row of the matrix represents the functional connectivity vector of a specific neuron with respect to all other neurons. Stability values for each neuron were then calculated using cosine similarity between adjacent functional connectivity vectors across the whole series of time bins. Individual neuronal stability was obtained as described for whole-network stability (above) by averaging cosine similarity values across each recording period.

To quantify how the strength of functional connectivity changed between neurons before versus after a period of stimulation (under both anaesthetized and non-anaesthetized conditions), we calculated the differences between corresponding elements of the functional connectivity matrix, using the AMD metric described above. To quantify the long-term effects of optogenetic stimulation on connection strength, we quantified pairwise connectivity for the last 15 min of baseline recording, and the last 15 min of a 2-h post-stimulation recovery

period. We then subtracted post-stimulation values from baseline. Distributions of these functional connectivity changes were then compared for experiments using ChR2- versus GFP-expressing mice (Fig. 8g; Supplementary Figs 15h and 16d).

**Data availability.** The datasets generated during and/or analysed during the current study are available from the corresponding author on reasonable request.

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

## Acknowledgements

Support for this research was provided by a Young Investigator Award from the Brain and Behavioral Research Foundation, an Alfred P. Sloan Foundation Fellowship, and the NIH (DP2MH104119 and R01EB018297). We are grateful to Sha Jiang and Nora Lashner for expert technical assistance with these studies, the laboratory of Dr Cathy Collins (Department of Molecular, Cellular, and Developmental Biology, University of Michigan) for microscopy assistance, and Christopher Broussard and Igor Belopolsky (Advocacy and Research Support, College of Literature, Science and Arts Information Technology, University of Michigan) for programming assistance.

## Author contributions

Experiments were designed by S.J.A., with input from N.O. and M.Z. N.O. carried out all animal experiments with assistance from S.S. and S.J.A. N.O., S.J.A. and S.S. carried out spike sorting and analyses of sleep states, field activity and basic firing properties (for example, firing rate and spike field coherence). Network connectivity, stability and functional similarity metrics were developed and implemented by J.W., S.M. and D.M., with assistance from M.Z., N.O. and S.J.A. wrote the manuscript, with assistance from M.Z.

## Additional information

**Competing interests:** The authors declare no competing financial interests.

DOI: 10.1038/ncomms16120    **OPEN**

# Erratum: Parvalbumin-expressing interneurons coordinate hippocampal network dynamics required for memory consolidation

Nicolette Ognjanovski, Samantha Schaeffer, Jiaxing Wu, Sima Mofakham, Daniel Maruyama, Michal Zochowski & Sara J. Aton

*Nature Communications* 8:15039 doi: 10.1038/ncomms15039 (2017); Published 6 Apr 2017; Updated 6 Jul 2017

The original version of the Supplementary Information attached to this Article did not contain Supplementary Figures 11–16 and Supplementary References. The HTML has now been updated to include a corrected version of the Supplementary Information file.

