## [Peer Review File · Nature Communications]

Reviewers' Comments:

Reviewer #1 (Remarks to the Author)

In this paper, Ognjanovski et al. record and manipulate neurons of the CA1 area of the hippocampal formation before and after single trial contextual fear conditioning. Pharmacogenetic inhibition of CA1 PV+ interneurons block conditioning-induced network changes reflected in oscillations and inter-neuronal correlation structure as well as fear memory consolidation. Rhythmic stimulation of PV+ neurons resulted in opposite network effects in anesthetized animals. The authors conclude that PV+ interneurons of the hippocampus play a critical role in driving memory formation.

The cellular mechanisms underlying memory formation are clearly of importance. The authors' approach of combining a classical behavioral assay with specific and precise opto- and pharmacogenetics tools as well as more advanced data analysis could yield significant inroads into this problem. However, some strong methodological and conceptual concerns have to be addressed before the interpretation and impact of the present results can be assessed.

Major points

1. FS means fast spiking, referring to firing rate. If the CA1 cells were classified based on spike width, as claimed, then they should be called NS - narrow spiking. Generally there is very little methodological detail about the recordings. How were these cells classified exactly? The authors should include a supplementary figure showing that NS and wide spiking cells were clearly separated. Then PV+ cells in other sentences are called PV+ FS cells, however, the manipulations were not specific to the FS subset of PV+ neurons (most but not all PV+ cells are FS). This blurs the difference between analyzing PUTATIVE PV-neurons and manipulating PV-expressing cells. This difference should be made clear at every point, because NS (or FS) cells can at best be called 'putative' PV-interneurons.

A related concern is regarding spike sorting. The details of verifying cluster separation are entirely missing. What were the cluster quality measures (average, SD, cutoff, ideally full distribution) like isolation distance or L-ratio? Were putative single neurons checked for auto-correlation violations? This is important because cluster quality can strongly affect correlation measures and it is technically challenging to record stably separate clusters for 24 hours in CA1 pyramidal layer, even with tetrodes, let alone stereotrodes.

2. Summary figures of coherence changes (Fig. 1e-f) are not very informative. What is the explanation for significant changes after sham conditioning in Fig. 1f? What are the effects on individual neurons underlying these bar graphs? What is the distribution of coherence changes?

The (relatively low) sample sizes are missing both from the main text and the figures (they are buried in the methods section).

3. How was freezing assessed and how was it differentiated from immobility?

4. The paper is lacking a detailed characterization of post-CFC behavior and oscillations. First, a supplementary figure demonstrating assignment of NREM, REM and awake segments would be necessary. Fig. S2 shows that the coherence changes were not due to changes in the cumulative length of the different sleep epochs. But were they due to changes in the overall quantity or quality of the oscillations (theta, delta, ripple) or more due to changes in the firing patterns of the recorded neurons? Was there a change in the structure of sleep segments (hypnogram)? E.g. the length / onset of the first NREM segment, which is known to be the most important for memory consolidation. Relatedly, what explains the choice of 6 hours? What was the time scale of the changes? Did they tail off before the 6 hours? Conversely, did they outlast 6 hours? Although not strictly necessary, it would be interesting to see what happens to coherence with respect to ripples that are also important for memory consolidation.

5. Regarding the impact of PV-inhibition on oscillations, the summary bar graphs potentially hide important details of the effects. Was there a change in amplitude, frequency, appearance, spectral profile or bout length distribution (fragmentation) of these oscillations?

6. The analyses of network stability and functional connectivity are an innovative way of probing changes in the inter-neuronal correlation structure. I still wonder whether more traditional methods based on cross-correlations would yield similar or different results. Also, there are some methodological questions to address. How many neurons were recorded? How was 'stably-recorded' defined? Is there a likely explanation for vehicle and CNO causing opposite effects as compared to CNO blocking the changes seen in the vehicle-injected animals? I find Fig.5e problematic because correlation seem to be driven by two outliers.

7. There is no summary figure for FSM (Fig.6). Without that there is no proof that the results hold on average.

8. There is a strong concern about the PV stimulation experiments. Unlike the CNO experiments, where CA1 specificity was achieved by virus injection, these experiments were carried out in PV-ChR2 neurons, where all PV cells express channelrhodopsin. It is well known that medial septal PV+ GABAergic neurons project to the hippocampus and have very strong impact on oscillations, synchrony, firing patterns and even CFM (see e.g. Boyce, Glasgow, Williams and Adamantidis in Science 13 May 2016). In the present experiments, septal PV+ axons were likely also stimulated. Therefore these experiments are only credible if they are repeated with CA1 virus injections in Pv-Cre mice. Also, these experiments were carried out under anesthesia. Are the results the same in drug-free animals? Also, it would be nice to see whether this PV stimulation protocol enhances behavioral memory (in virus injected Pv-Cre mice). This is necessary if the authors would like to maintain their conclusion about the critical role of PV neurons in memory formation. Otherwise the conclusion should be modify to clearly only claim necessity without information on sufficiency.

In light of these comments, the conclusion sound too far-reaching in general. Based on the inactivation results, PV+ interneurons may be necessary for fear memory consolidation. However, as one of the major inhibitory components of the hippocampal network, they may only have an enabling or even a very indirect effect. Even if the optogenetic activation results prove to be specific, there is a lingering concern about the actual mechanistic contribution of this cell type. I therefore suggest toning down the conclusions accordingly.

Minor points

1. c-fos positivity of the top neuron in Fig.2c is not very convincing. Was there a mechanism to account for classification errors of the experimenter? Is there a better example for this figure?
2. It is hard to see the baselines in Fig 4a-b but they seem to be different for CNO and control. Is there an explanation? Fig 4c - why is control-CNO different for hM4Di-vehicle?
3. Fig. 4 - All sample sizes are missing.
4. Fig 4i - What was significant here?
5. Fig. 5d - Please justify the choice of t-test whereas non-parametric tests are applied in other figures.
6. Fig 6 legend - Reference to Fig 4b is prob. Fig. 5b.
7. Fig. 7a - This figure is not very informative. Top left raster is pitch black. I assume the stimulation rasters look "rhythmic" because they are aligned to each pulse in the 8Hz pulse train. That is showing the same information multiple times compressed into a small figure - it would be more informative to align to the first pulse only. I don't understand why the baseline rasters look "rhythmic", too.
8. Fig 7b, right - I could not find information on these panels in the legend.
9. Fig. 7c - Is this during or after stimulation?
10. Functional connectivity analysis - Lines 432-434 are unclear to me. What is L exactly, and how were the moments calculated?
11. Consider discussing the related papers "Parvalbumin-expressing basket-cell network plasticity induced by experience regulates adult learning" by Donato, Rompani and Caroni and "Identification of Parvalbumin Interneurons as Cellular Substrate of Fear Memory Persistence" by Caliskan et al. (Cerebral Cortex).

Reviewer #2 (Remarks to the Author)

This manuscript demonstrates that following contextual fear conditioning putative excitatory and inhibitory multiunit activity showed enhanced coherence with low frequency local field potentials (LFP) both and waking and sleep states. DREADD-induced suppression of parvalbumin expressing interneurons following the fear conditioning prevented the consolidation of fear memories and reduced the theta and delta-band LFP power. They also found that the DREADD-suppression also reduced the stable, coherent activation patterns of the recorded units. In anesthesia, the rhythmic activation of the parvalbumin expressing interneurons resulted in an

increase in the stability of coherent activation of the recorded units even after the stimulation. Overall the study contains important, novel findings. The weakness of the study is that it did not follow the same technical standards as the 'place cell field' and, because of it, some of the conclusions are indirect. I have several technical questions, some may help to draw more direct conclusions.

Major comments

In using stereotrodes, it is impossible to reliably isolate single units. But, because of the waveshape (i.e. width) differences, it is possible to isolate the multiunit activity of interneurons and pyramidal cells. It should be acknowledged. It should be also discussed whether the >48h tethered recordings may have partially biased the results.

In the waking state it would be important to differentiate active and inactive waking states. If the animals' speed was not recorded, at least try to separate theta oscillatory periods and non-theta periods. For example it would be important to know whether interneurons show similar (opposite to the usual) trend during theta oscillations in active waking and REM periods. Note also that delta power during REM sleep is not very meaningful; in practice pure REM sleep associated with theta oscillations should not contain any delta waves.

Figure 4 is interesting but it is impossible to differentiate the curves in a,b, g and h. g and h seem to contain a grey curve as well not labeled on the figure. In h it is also not clear why the baseline does not show increased ripple power at all. This should contain ripples. In c and e the vehicle control case is missing and the labeling is different in a & b compared to c and e.

The functional connectivity analysis (Fig. 5) is interesting but it is hard to interpret. There is a concern that stability of these connectivity patterns is not seen in the CNO case because ripples are suppressed and consecutive 1min time windows may not contain sufficient number of sharp waves to estimate average connectivity of the network. Would this effect hold if longer time periods are used, e.g. , 10sec?

Fig 6 shows interesting examples but would need some quantification (over different recordings and animals) or otherwise please remove the figure. The two baselines shown in b look quite different and the reader may wonder why?

Minor

Note that parvalbumin expression is not able to differentiate perisomatic innervating

interneurons, it is also expressed in dendritic-targeting interneurons, e.g., bistratified cells. Please take this into consideration in revising the manuscript.

Response to Reviewers

Reviewers' comments:

Reviewer #1 (Remarks to the Author):

In this paper, Ognjanovski et al. record and manipulate neurons of the CA1 area of the hippocampal formation before and after single trial contextual fear conditioning. Pharmacogenetic inhibition of CA1 PV+ interneurons block conditioning-induced network changes reflected in oscillations and inter-neuronal correlation structure as well as fear memory consolidation. Rhythmic stimulation of PV+ neurons resulted in opposite network effects in anesthetized animals. The authors conclude that PV+ interneurons of the hippocampus play a critical role in driving memory formation.

The cellular mechanisms underlying memory formation are clearly of importance. The authors' approach of combining a classical behavioral assay with specific and precise opto- and pharmacogenetics tools as well as more advanced data analysis could yield significant inroads into this problem. However, some strong methodological and conceptual concerns have to be addressed before the interpretation and impact of the present results can be assessed.

Major points

1. FS means fast spiking, referring to firing rate. If the CA1 cells were classified based on spike width, as claimed, then they should be called NS - narrow spiking. Generally there is very little methodological detail about the recordings. How were these cells classified exactly? The authors should include a supplementary figure showing that NS and wide spiking cells were clearly separated. Then PV+ cells in other sentences are called PV+ FS cells, however, the manipulations were not specific to the FS subset of PV+ neurons (most but not all PV+ cells are FS). This blurs the difference between analyzing PUTATIVE PV-neurons and manipulating PV-expressing cells. This difference should be made clear at every point, because NS (or FS) cells can at best be called 'putative' PV-interneurons.

We agree with the reviewer that from a technical standpoint, “narrow-spiking” more accurately describes criteria used by researchers (including ourselves) to identify these interneurons electrophysiologically. However, “fast-spiking” is conventionally used in the field to refer to (typically PV-expressing) interneurons with short-duration waveforms. For example, a Pubmed search for “fast-spiking”+interneurons+parvalbumin yields 243 results, while “narrow-spiking”+interneurons+parvalbumin yields 0 results. A simpler search for “fast-spiking”+interneurons yields 703 results, while “narrow-spiking”+interneurons yields 10 results. Thus despite the fact that spike rate *per se* was not used as a classification criterion, for the sake of using generally-accepted terminology we feel that the term “fast-spiking” should be retained. We have clarified the criteria we used for fast-spiking interneuron classification in the new **Supplementary Fig. 1b**. Here we show distributions of spike half-widths for all neurons recorded, and the cutoff for classification as a FS interneuron. We have also outlined these criteria in the revised **Methods** section.

The issue of PV expression within this population is another important point, and we appreciate the reviewer's concern. For simplicity, we have eliminated references to "PV+ FS interneurons". We now refer to "FS interneurons" only in cases where we specifically classified cells as fast-spiking using electrophysiological criteria. For all other experiments (*i.e.*, those involving pharmacogenetic and optogenetic manipulations) we refer simply to "PV+ interneurons".

A related concern is regarding spike sorting. The details of verifying cluster separation are entirely missing. What were the cluster quality measures (average, SD, cutoff, ideally full distribution) like isolation distance or L-ratio? Were putative single neurons checked for auto-correlation violations? This is important because cluster quality can strongly affect correlation measures and it is technically challenging to record stably separate clusters for 24 hours in CA1 pyramidal layer, even with tetrodes, let alone stereotrodes.

We appreciate the reviewer's concern, and agree that stereotrode or tetrode recording of single neurons > 24 h duration is a technical challenge. While relatively few labs attempt to do such recordings, our lab routinely carries out recordings of up to 48 h (the duration reported in this manuscript). In support of our ability to consistently identify and separate spike clusters, we now provide: 1) **Supplementary Fig. 1a**, which shows an example of spike clusters from two neurons continuously recorded over 48 h, and 2) more technical information in the **Methods** section regarding cluster quality metrics used to validate cluster isolation. We now specify that we eliminated clusters with interspike interval (ISI)-based absolute refractory period violations, reducing the likelihood of including multiunit data. We also provide data for cluster separation (MANOVA and Davies-Bouldin [DB] validity index; **Supplementary Fig. 1c**), which compare favorably with the same measurements reported by others for isolated single-unit data.

Only those neurons which could be reliably tracked and discriminated, based numerous criteria (including appropriate cluster separation, ISI-based discrimination metrics, consistency of spike waveforms across time, etc.) were included in our analyses. Due to the stringency of our criteria for inclusion, many neurons' spikes that either were lost or appeared during recording were eliminated from the analyses presented. On the other hand, that data which was included is very likely reflecting the activity of single neurons (not multiunit activity). Of the total number of recording channels with data included in the study, 137 (88%) contributed only one stably-recorded single unit, 16 contributed two units, and only 2 contributed three units. Thus the stringency of our inclusion criteria itself limits the degree to which ineffective cluster separation could feasibly affect our results.

2. Summary figures of coherence changes (Fig. 1e-f) are not very informative. What is the explanation for significant changes after sham conditioning in Fig. 1f? What are the effects on individual neurons underlying these bar graphs? What is the distribution of coherence changes? The (relatively low) sample sizes are missing both from the main text and the figures (they are buried in the methods section).

We thank the reviewer for pointing out that we omitted the number of neurons that went into our analysis in **Fig. 1** - we have revised the figure to include this information for both cell types and groups. Instead of presenting the mean of firing coherence changes, which did not capture the distribution of these changes, we now show percentile boxplots with individual data points. We hope that this makes the range (i.e., variability) and magnitude of changes that we see across individual neurons more clear.

3. How was freezing assessed and how was it differentiated from immobility?

This is an important question, as the definition of this behavior is critical to how we interpret the effects of our pharmacogenetic manipulations. We used standard criteria in the field to define freezing: *i.e.*, crouched posture with a lack of all movement (including head or whisker movement) save respiration, as described previously (Fanselow 1986). We have added a more-detailed description of criteria for scoring freezing behavior in our **Methods** section.

4. The paper is lacking a detailed characterization of post-CFC behavior and oscillations. First, a supplementary figure demonstrating assignment of NREM, REM and awake segments would be necessary. Fig. S2 shows that the coherence changes were not due to changes in the cumulative length of the different sleep epochs. But were they due to changes in the overall quantity or quality of the oscillations (theta, delta, ripple) or more due to changes in the firing patterns of the recorded neurons? Was there a change in the structure of sleep segments (hypnogram)? E.g. the length / onset of the first NREM segment, which is known to be the most important for memory consolidation. Relatedly, what explains the choice of 6 hours? What was the time scale of the changes? Did they tail off before the 6 hours? Conversely, did they outlast 6 hours? Although not strictly necessary, it would be interesting to see what happens to coherence with respect to ripples that are also important for memory Consolidation.

We agree with the reviewer that more information can be provided to clarify how we classified post-CFC behavioral states. In response to the reviewer's first point, we now provide **Supplementary Fig. 3a**, which shows differences between NREM, REM, and wake states based on LFP and EMG parameters. This illustrates our basis for categorization of behavioral states. We also now include a more clear description of how intervals of recording were classified as NREM, REM or wake in the **Methods** section.

In response to the second point, we have quantified additional parameters of sleep architecture in both wild-type and *Pvalb-IRES-CRE* transgenic mice (**Supplementary Figs. 2 and 4**, respectively). These include: 1) time to the first bout of REM, NREM, or wake, 2) duration of first bout of REM, NREM, or wake, and 3) hypnograms showing transitions between states (in representative animals) over the first 6 h of baseline and post-conditioning recording. Based on the lack of significant differences in sleep architecture when comparing these additional

variables, we feel that changes in sleep itself do not explain the changes in sleep-associated oscillations that we see.

The question of whether the changes we see in spike-field coherence are due to a change in oscillatory dynamics or firing pattern is an interesting one. We feel that a parsimonious explanation is that both are contributing. We see clear changes in the LFP with regard to the nature of oscillations (with amplitude and/or frequency of occurrence increasing after CFC in various frequency ranges). Based on this, we conclude that oscillatory dynamics must be changing. If neuronal firing were random with respect to these oscillations, we would not expect to see increasingly coherent firing during this time. Because we do see changes in firing pattern as well, it seems likely that *both* firing and oscillatory dynamics are changing in a coordinated fashion.

We acknowledge that our rationale for focusing on the first 6 h post-training was unclear as previously written. We are grateful to the reviewer for pointing out this out, as this feedback was helpful for revising the manuscript. There are three reasons that we have emphasized changes occurring over the first 6 h. First, prior work has shown that sleep within the first 5-6 h after CFC is essential for consolidating CFM; sleep deprivation after this time window does not disrupt fear memory (Graves *et al.*, 2003). Second, based on prior work, we anticipate that our pharmacogenetic manipulation of PV+ interneuron activity (initiated by systemic administration of CNO immediately following CFC) will occur primarily in this 6-h window. Third, we see that CFC-induced changes in power spectral density seen in control mice are largely absent after 6 h. We now show the change in spectral power from baseline over hours 6-12 post-CFC (from baseline at the same time period) in **Supplementary Fig. 5b**.

Finally, the reviewer makes an excellent point about ripple firing coherence. We were unable to detect a significant change in ripple spike-field coherence in our wild-type mouse recordings. However, we did find both a significant increase in the occurrence of ripples in these recordings (mirroring findings from pharmacogenetic experiments in *Pvalb-IRES-CRE* mice), and also significant changes in the frequency of neuronal firing during these oscillations. We now present these data in our revised **Fig. 1**.

5. Regarding the impact of PV-inhibition on oscillations, the summary bar graphs potentially hide important details of the effects. Was there a change in amplitude, frequency, appearance, spectral profile or bout length distribution (fragmentation) of these oscillations?

We agree that simply showing summary PSDs (i.e., those showing average spectral power across 6 h) and bar graphs showing integrated power changes across a frequency band does not allow a nuanced interpretation of how these oscillations change over time. To better capture some of these details, we now show spectrograms for representative LFPs from individual mice expressing hM4Di, following post-CFC administration of either CNO or vehicle. We feel that this

provides a better general picture of changes in spectral power occurring across individual bouts of wake, NREM and REM.

6. The analyses of network stability and functional connectivity are an innovative way of probing changes in the inter-neuronal correlation structure. I still wonder whether more traditional methods based on cross-correlations would yield similar or different results.

We agree that the question of whether AMD-based functional connectivity metrics and traditional cross-correlation yield similar results (*i.e.*, for network stability) is both interesting and very important. Like cross-correlation, the AMD metric (introduced initially in Feldt *et al.*, 2009) can provide an estimate of statistical significance for co-firing between two neurons. Unlike cross-correlation, AMD does not require that ISIs between the neurons be nearly identical from spike pair to spike pair. Instead, it relies on the ISI relationship between two neurons being significantly shorter than expected by chance (based on each neuron's individual ISI distribution; see **Methods** and **Fig. 6a**). This provides a weaker but adequate constraint, which we believe is crucial for viable estimation of functional connectivity in noisy environments (like the brain of a behaving animal).

Supplementary Fig. 9a and **b** provide an illustration of performance of AMD vs. cross-correlation on 1) two simulated spike trains, and 2) an *in vivo* data set. We see that AMD-based functional connectivity provides a more reliable estimate of spike train coincidence than cross-correlation does. AMD provides the following additional advantages when compared with cross-correlation:

- Cross-correlation has two free parameters: 1) the time shift (*i.e.*, delay) between the spike trains and 2) the size of Gaussian convolution of spike timings (see **Supplementary Fig. 9**). While there are rules to constrain the Gaussian used for convolution, we show that the results vary significantly as a function of this parameter. The time delay parameter constitutes another problem. While for two neurons' spike trains, one can use a specific delay for which cross-correlation is maximized, this information 1) cannot be applied to other neuronal pairs in the network and 2) may change for the same neuronal pair across time. Thus while cross correlation is good for detecting exact pattern repetitions among an ensemble of neurons, it will not be informative if the pattern is shifting over time, or noisy.
- AMD significance is estimated analytically; thus the computation is rapid. In contrast, the bootstrapping required for calculating cross-correlation significance slows the calculation dramatically. This is especially important when one is calculating changes in the pattern of functional connectivity (as we do here to compute network stability) over thousands of time windows, as we do here.

Supplementary Fig. 9b compares application of AMD and cross-correlation metrics to calculation of network stability (across NREM sleep) for a representative mouse. We generate FSMs using the two techniques at baseline and after CFC. Using cross-correlation, we see that

the network looks similar during the two recording periods (*i.e.*, there is little change after CFC). In contrast, network similarity calculated using AMD-based metrics reveals a significant shift towards higher stability after CFC. Overall mean similarity reported by cross-correlation is also somewhat lower than that reported using AMD. Taken together, these data suggest that AMD-based metrics are not only more efficient from a computational standpoint, they are also potentially both more sensitive and reliable for detecting changes in network dynamics in response to perturbations such as single-trial learning.

Also, there are some methodological questions to address. How many neurons were recorded? How was 'stably-recorded' defined? Is there a likely explanation for vehicle and CNO causing opposite effects as compared to CNO blocking the changes seen in the vehicle-injected animals? I find Fig.5e problematic because correlation seem to be driven by two outliers.

We have made an effort to more clearly describe our definition of “stably-recorded”, in response to the reviewer’s earlier point. We also have reported the range of *ns* for stably-recorded neurons from each mouse, which were used for these calculations.

The issue of decreases in network stability (relative to baseline) seen in CNO-treated mice is also important. Our interpretation of this result is that with reduced PV+ interneuron activity, stability is reduced relative to baseline, which means that at baseline, stability of functional connectivity is maintained at least in part through the activity of this cell population. We have revised our discussion section to reflect this interpretation.

We thank the reviewer for pointing out that some features of the methodology were unclear. We hope that by defining what we mean by stably-recorded neurons (in response to the reviewer’s first point, above) we have clarified our inclusion criteria. We have also revised the figure legend corresponding to this analysis to give the range of the number of neurons recorded per animal.

We agree with the reviewer’s concern regarding the correlation in **Fig. 5e**. Because each animal provided only one data point for this analysis, assessing this relationship is necessarily limited by the *n*. To provide greater resolution, and because the reviewer also suggested that we characterize ripple oscillations in wild-type mice (which we have done, in response to point 4 above), we calculated ripple frequency vs. NREM stability relationships for these mice as well. We found that these data corroborated the relationship between ripple occurrence and network stability. This was true for mean ripple frequency over the first 6 h following training ($R = 0.56$, $p < 0.05$, Spearman rank order). However, a stronger relationship was found between NREM stability and ripple frequency in the first 2 h following training (where the largest increases from baseline are seen after CFC across groups); this relationship is now shown in revised **Fig. 6e**.

7. There is no summary figure for FSM (Fig.6). Without that there is no proof that the results hold on average.

We are grateful to the reviewer for pointing this out. We have now quantified FSM similarity scores over time, and show distributions of minute-to-minute similarity values both within (**Fig. 7c** and **Supplementary Fig. 9b**) and across (**Fig. 7d**) individual mouse recordings. In all cases, we see statistically significant shifts in the distribution of similarity values toward greater similarity in vehicle-treated mice (with intact memory consolidation). This shift is never seen in CNO-treated mice (with impaired consolidation).

8. There is a strong concern about the PV stimulation experiments. Unlike the CNO experiments, where CA1 specificity was achieved by virus injection, these experiments were carried out in PV-ChR2 neurons, where all PV cells express channelrhodopsin. It is well known that medial septal PV+ GABAergic neurons project to the hippocampus and have very strong impact on oscillations, synchrony, firing patterns and even CFM (see e.g. Boyce, Glasgow, Williams and Adamantidis in Science 13 May 2016). In the present experiments, septal PV+ axons were likely also stimulated. Therefore these experiments are only credible if they are repeated with CA1 virus injections in Pv-Cre mice. Also, these experiments were carried out under anesthesia. Are the results the same in drug-free animals? Also, it would be nice to see whether this PV stimulation protocol enhances behavioral memory (in virus injected Pv-Cre mice). This is necessary if the authors would like to maintain their conclusion about the critical role of PV neurons in memory formation. Otherwise the conclusion should be modify to clearly only claim necessity without information on sufficiency.

We appreciate the reviewer's concern regarding the potential effects of expressing (and activating) ChR2 in non-CA1 PV+ interneurons. To address this concern, we have repeated our acute recording experiments using *Pvalb-IRES-CRE* mice with AAV-mediated, CA1-specific transduction of ChR2. Using the same stimulation parameters, we find that: 1) stimulation of CA1 PV+ interneurons over a range of frequencies (but particularly 4-10 Hz) leads to a significant increase in spike field coherence across the population of CA1 neurons, 2) within individual mice, CA1 neurons synchronize to network oscillations with different phase angles of entrainment (*i.e.*, some fire in phase with stimulation, some fire in antiphase), 3) neurons' increases in spike field coherence correlate with increases in functional connectivity stability, 4) stimulation at a mid-range theta frequency (7 Hz) causes a long-lasting increase in network stability (measurable 2 h post-stimulation), and 5) 7 Hz stimulation also causes a long-lasting increase in functional connections' strength across the network. We present these new data in **Supplementary Figs. 14** and **15**. Because these effects are nearly identical to those we initially reported for *PV:ChR2* transgenic mice, we conclude that they are best explained by *local* effects of CA1 PV+ interneurons on the CA1 network.

In light of these comments, the conclusion sound too far-reaching in general. Based on the inactivation results, PV+ interneurons may be necessary for fear memory consolidation. However, as one of the major inhibitory components of the hippocampal network, they may only

have an enabling or even a very indirect effect. Even if the optogenetic activation results prove to be specific, there is a lingering concern about the actual mechanistic contribution of this cell type. I therefore suggest toning down the conclusions accordingly.

We appreciate the reviewer's concern, based on the issues raised in review. Having carried out the additional optogenetic experiments to isolate effects within CA1, and having carried out additional analyses of the effects of inactivation on CA1 network dynamics, we feel more confident in our assertion that PV interneurons pattern local CA1 activity in the context of memory consolidation. However, we have revised the discussion to provide a more cautious interpretation of our present data.

Minor points

1. c-fos positivity of the top neuron in Fig.2c is not very convincing. Was there a mechanism to account for classification errors of the experimenter? Is there a better example for this figure?

We agree with the reviewer that the image shown was not as clear as it should have been, and thank them for pointing this out. A new image has been provided for **Fig. 2**, which shows clearer c-Fos expression for the PV-expressing neurons in the field of view.

Images were collected for all brain slices containing virally-transduced CA1 and *post hoc* analysis of c-Fos immunoreactivity was conducted with the scorer blind to hemisphere. Additional details for quantification of c-fos-positive and -negative neurons is now provided in Methods section.

2. It is hard to see the baselines in Fig 4a-b but they seem to be different for CNO and control. Is there an explanation? Fig 4c - why is control-CNO different for hM4Di-vehicle?

We are grateful to the reviewer for pointing out the difficulty in discriminating the two baseline power spectra in **Fig. 4**. Baseline power spectra may look different from animal to animal, and even for the same animal there may be slight differences between baseline recording periods occurring weeks apart. However, the differences in **Fig. a-b** were relatively minor - it was just difficult to distinguish them, as the reviewer noted. For clarity, we have now separated out the CNO and DMSO treatment data in **Supplementary Fig. 5**.

We feel that the changes in delta frequency activity during REM (**Fig. 4 c**) may actually reflect a harmonic of theta, which tends to change in parallel with changes in 4-8 Hz activity. While we do not have a clear explanation of why this value would be different between control-CNO and hM4Di-vehicle, we do not feel that REM-associated changes in this particular frequency band necessarily reflect the dominant oscillation occurring in CA1 during this state.

3. *Fig. 4 - All sample sizes are missing.*

We thank the reviewer for pointing this out. We have amended the figure legends to reflect the number of LFP channels included in these analyses.

4. *Fig 4i - What was significant here?*

We are grateful to the reviewer for pointing out this omission on our part. Asterisks and significance bars were added to reflect pair-wise comparisons.

5. *Fig. 5d - Please justify the choice of t-test whereas non-parametric tests are applied in other figures.*

We have replaced this with a non-parametric test, with the same statistical outcome.

6. *Fig 6 legend - Reference to Fig 4b is prob. Fig. 5b.*

This is correct. We thank the reviewer for pointing out this typographical error, which we have corrected.

7. *Fig. 7a - This figure is not very informative. Top left raster is pitch black. I assume the stimulation rasters look "rhythmic" because they are aligned to each pulse in the 8Hz pulse train. That is showing the same information multiple times compressed into a small figure - it would be more informative to align to the first pulse only. I don't understand why the baseline rasters look "rhythmic", too.*

We have remade these spike rasters to allow more "white space", and thus better discrimination of individual spikes for fast spiking neurons. Because the light pulses are delivered rhythmically, it is not possible to align rasters to the first pulse, without subsequent pulses showing up later in the 1-s raster. It is true that data are multiply plotted to show spikes with reference to every pulse, this is typical in many fields where biological rhythms are shown in register with a periodic stimulus - e.g., "actograms" in the circadian rhythms field. While this may emphasize the rhythmic nature of the spike-light pulse relationship, it is not creating an artifact in the data. We hope that the cumulative histograms of firing shown under each spike raster illustrates that the rhythmicity of firing (i.e., minimum vs. maximum firing in a given temporal window) is changing as a function of stimulation.

Baseline rasters (for both ChR2 and GFP-expressing mice) look somewhat “rhythmic” as well because, for the sake of fair comparison with stimulation periods, a “fictive” rhythmic light pulse train is added to the data file (taken from a file where rhythmic light stimulation occurs), and rasters are plotted with respect to the fictive light pulse events. This indicates that even at baseline, firing shows some relationship to a (for example) 8-Hz signal. We hope that this also addresses the reviewer’s first point, regarding the effects of a rhythmic reference signal on rhythmic synchrony of firing.

8. Fig 7b, right - I could not find information on these panels in the legend.

We thank the reviewer for pointing this out - we now describe what the right-hand panels in 7b (now **Fig. 8b**) contain.

9. Fig. 7c - Is this during or after stimulation?

This is during rhythmic stimulation. The text in the figure legend has been amended to clarify this.

10. Functional connectivity analysis - Lines 432-434 are unclear to me. What is L exactly, and how were the moments calculated?

We have clarified the **Methods** section to clarify both that L refers to the ISI of the reference spike train, and how the moments were calculated.

11. Consider discussing the related papers "Parvalbumin-expressing basket-cell network plasticity induced by experience regulates adult learning" by Donato, Rompani and Caroni and "Identification of Parvalbumin Interneurons as Cellular Substrate of Fear Memory Persistence" by Caliskan et al. (Cerebral Cortex).

We are grateful to the reviewer for bringing these articles to our attention, as they suggest a potential link between network activity changes mediated by PV+ interneurons and plasticity in this cell population. We now discuss our findings in light of these studies.

Reviewer #2 (Remarks to the Author):

This manuscript demonstrates that following contextual fear conditioning putative excitatory and inhibitory multiunit activity showed enhanced coherence with low frequency local field potentials (LFP) both and waking and sleep states. DREADD-induced suppression of parvalbumin

expressing interneurons following the fear conditioning prevented the consolidation of fear memories and reduced the theta and delta-band LFP power. They also found that the DREADD-suppression also reduced the stable, coherent activation patterns of the recorded units. In anesthesia, the rhythmic activation of the parvalbumin expressing interneurons resulted in an increase in the stability of coherent activation of the recorded units even after the stimulation. Overall the study contains important, novel findings. The weakness of the study is that it did not follow the same technical standards as the 'place cell field' and, because of it, some of the conclusions are indirect. I have several technical questions, some may help to draw more direct conclusions.

Major comments

In using stereotrodes, it is impossible to reliably isolate single units. But, because of the waveshape (i.e. width) differences, it is possible to isolate the multiunit activity of interneurons and pyramidal cells. It should be acknowledged. It should be also discussed whether the >48h tethered recordings may have partially biased the results.

It is true that there are differences between the number of single units that can be reliably discriminated on tetrodes vs. stereotrodes (with tetrodes discriminating more single units per site in areas where neuronal cell bodies are tightly packed - such as the pyramidal cell layer in CA1). However, it is not impossible to reliably isolate single units using stereotrodes (based on published data from both our lab and others), and our laboratory routinely uses stereotrodes to isolate single unit activity in freely-behaving animals. In support of our ability to consistently identify and separate spike clusters, we now provide: 1) **Supplementary Fig. 1a**, which shows an example of spike clusters from two neurons continuously recorded over 48 h, and 2) more technical information in the **Methods** section regarding cluster quality metrics used to validate cluster isolation. We now specify that we eliminated clusters with interspike interval (ISI)-based absolute refractory period violations, reducing the likelihood of including multiunit data. We also provide data for cluster separation (MANOVA and Davies-Bouldin [DB] validity index; **Supplementary Fig. 1c**), which compare favorably with the same measurements reported by others for isolated single-unit data.

Only those neurons which could be reliably tracked and discriminated, based numerous criteria (including appropriate cluster separation, ISI-based discrimination metrics, consistency of spike waveforms across time, etc.) were included in our analyses. Thus many neurons' spikes that either were lost or appeared during recording were eliminated from our data, but that data which was included is very likely reflecting the activity of single neurons (not multiunit activity). Of the total number of recording channels with data included in the study, 137 (88%) contributed only one stably-recorded single unit, 16 contributed two units, and only 2 contributed three units. Thus the stringency of our inclusion criteria itself limits the degree to which ineffective cluster separation could affect our results.

As the reviewer points out, our experimental procedures require continuous recording of neural activity for long periods of time (~48 h), and is possible that tethering itself may lead to

alterations in animals' behavior. Our pre-recording handling and habituation procedures are aimed at minimizing these potential effects. We have revised the **Methods** section text to clarify that all mice are habituated to all features of the recording apparatus, including tethering via lightweight cables designed for use in mice, prior to recordings. Because the duration of tethering is similar for all behavioral studies reported here, we do not anticipate that differences between experimental groups can be attributed to the recording apparatus itself.

In the waking state it would be important to differentiate active and inactive waking states. If the animals' speed was not recorded, at least try to separate theta oscillatory periods and non-theta periods. For example it would be important to know whether interneurons show similar (opposite to the usual) trend during theta oscillations in active waking and REM periods. Note also that delta power during REM sleep is not very meaningful; in practice pure REM sleep associated with theta oscillations should not contain any delta waves.

We thank the reviewer for pointing out that we did not fully address changes in spectral power during wake. We now show (in **Supplementary Fig. 5**) quantification of delta and theta power during wake, including representative LFP PSD changes and a summary of integrated power changes in these two bands. Because we see reductions in wake theta following CFC in vehicle-treated mice (and in CNO-treated mice expressing a control vector), a parsimonious explanation is that post-CFC changes may be different in wake than in REM. However, because (as the reviewer points out) wake theta power can be affected in a number of ways (including based on the animal's activity), and because we do see increases in *theta-frequency spike-field coherence* in wake (that are similar to changes in REM; **Fig. 1**), we do not feel we have sufficient grounds to speculate with regard to the cause of this.

Our interpretation of the changes we see in REM delta power is that this likely reflects a harmonic of the strong theta-frequency peak, rather than a discrete change in delta oscillations that occurs independently of theta changes.

Figure 4 is interesting but it is impossible to differentiate the curves in a,b, g and h. g and h seem to contain a grey curve as well not labeled on the figure. In h it is also not clear why the baseline does not show increased ripple power at all. This should contain ripples. In c and e the vehicle control case is missing and the labeling is different in a & b compared to c and e.

We thank the reviewer for pointing out that these curves were difficult to discriminate. We have modified the axes and the density of data points for the PSD data in the figure, which we hope makes it easier to differentiate. Some of the difficulty with panels a and b is due to the fact that spectral power baseline values (on both experimental days) and values after CFC with CNO treatment show a great deal of overlap, and are dwarfed by power changes after CFC on vehicle treatment days. To help discern differences between these curves, we have added

Supplementary Fig. 5a, which shows CNO and vehicle treatments separately for the same animals (and also includes values for wake, in response to the previous point).

Regarding the ripple power at baseline in this figure (now revised **Fig. 5**), there is a peak in power at baseline in these recordings at ~175 Hz, but the increase in peak power seen after CFC dwarfs this peak.

The functional connectivity analysis (Fig. 5) is interesting but it is hard to interpret. There is a concern that stability of these connectivity patterns is not seen in the CNO case because ripples are suppressed and consecutive 1min time windows may not contain sufficient number of sharp waves to estimate average connectivity of the network. Would this effect hold if longer time periods are used, e.g., 10sec?

The issue of decreases in network stability (relative to baseline) seen in CNO-treated mice is very important. Our interpretation of this result is that with reduced PV+ interneuron activity, stability is reduced relative to baseline, which means that at baseline, stability of functional connectivity is maintained at least in part through the activity of this cell population. This may be due to the role this cell population plays in promoting ripple oscillations, or through some other means (i.e., by regularizing firing patterns across the network generally, through inhibition).

The fact that post-training stability is tightly correlated with rate of ripple events in CA1 (see revised **Fig. 6**) suggests that having a sufficient number of ripples might be critical for maintaining stability. We agree that there is a strong possibility that stability deficits after CNO could be due to suppression of ripples (and perhaps also due to suppression of other oscillations like theta and delta). To test whether increasing the number of these oscillations per unit of recording (i.e., using periods greater than 1 min to calculate AMD connectivity) alters network stability outcomes, we recalculated AMD using a larger time window. The number of 10 minute long time windows containing continuous NREM or wake in any given recording was necessarily limited; this is expected due to the relatively fragmented sleep architecture typical of mice. For simplicity, we calculated AMD-based network connectivity patterns using the 5 longest continuous periods of NREM occurring in the (24-h) baseline or post-CFC periods. We then calculated stability across these 5 time periods. FSMs for these data (**Supplementary Fig. 10b**) indicate that there may indeed be greater stability post-CFC in the CNO condition *when only the longest epochs of NREM are considered*. Thus, we cannot rule out the possibility that stability is linked closely to the occurrence of ripples in NREM sleep.

Fig 6 shows interesting examples but would need some quantification (over different recordings and animals) or otherwise please remove the figure. The two baselines shown in b look quite different and the reader may wonder why?

We are grateful to the reviewer for pointing this out. We have now quantified FSM similarity scores over time, and show distributions of minute-to-minute similarity values both within (**Fig. 7c** and **Supplementary Fig. 9b**) and across (**Fig. 7d**) individual mouse recordings. In all cases, we see statistically significant shifts in the distribution of similarity values toward greater similarity in vehicle-treated mice (with intact memory consolidation). This shift is never seen in CNO-treated mice (with impaired consolidation).

Baseline network stability and FSMs look different from animal to animal, and even for the same animal for different baseline recording periods occurring weeks apart. We expect this is due to the fact that we are sampling different CA1 neurons with different baseline functional connectivity patterns. For this reason, in our network stability analyses we focus exclusively on post-CFC changes from baseline recording (and only for those neurons which are stably recorded across the entire baseline and post-CFC periods), rather than quantifying absolute values in different mice and experimental conditions.

Minor

Note that parvalbumin expression is not able to differentiate perisomatic innervating interneurons, it is also expressed in dendritic-targeting interneurons, e.g., bistratified cells. Please take this into consideration in revising the manuscript.

We are grateful for this feedback, which raises a very important point. To avoid confusion on this point, we have eliminated references to perisomatic innervation as an essential part of the mechanism for PV+ interneuron-mediated network regulation.

Reviewers' Comments:

Reviewer #1 (Remarks to the Author):

Overall, the authors were responsive to many of my points and the manuscript has significantly improved. I was especially impressed by the authors showing that the PV stimulation results are independent of septal inputs and by the analysis of ripples. However, I have some remaining concerns (see below in details). Beyond the methodological concern of cluster quality,

- I find that the results of PV inhibition on theta and delta oscillations are not entirely convincing.

- there is the lingering concern about the significance of optogenetic PV stimulation, as it was performed under anesthesia and without behavioral correlates.

The numbers refer to my original points.

1. Still, a random check of papers from the Pubmed search the authors referred to showed that it is uncommon to study “fast-spiking” interneurons without any quantification of their firing rates. Identification procedures often include double criteria based on spike width and firing rate. So as a minimum, the firing rate distribution of “FS interneurons” in this study would be a valuable addition.

Note that cluster quality does not only depend on separation of cluster pairs, but also separation of a putative single neuron cluster from everything else (including the so-called “noise cluster”). In that effect, Fig. S1c provides very limited information on cluster quality.

The analysis of ripple in Fig.1f-g is a good addition.

5. The representative LFP spectrograms do not really make up for the lack of thorough analysis I raised in my comment. In fact, they rather suggest that the mean effect could have been a result of rather sporadic events than typical oscillatory behavior, which diminishes my enthusiasm about this result. This is strengthened by the answer to minor point 2 - it is still unclear why control-CNO differs from vehicle. A harmonic is always a positive integer multiplier of the base frequency, so theta harmonics appear above the theta range.

6. The comparison of AMD to CCG is a nice addition; however, Fig. S9a is quite unclear.

8. I appreciate the demonstration of CA1-specific PV-stimulation. Nevertheless, this as superior experimental design compared to the problematic original one, therefore it should be presented in

the main figure and not in supplement.

minor point 7. When the same data is presented repeatedly in a rhythmic fashion - like in these raster plots - there is an artificial rhythmicity. This is very easy to demonstrate and indeed the authors themselves do the demonstration: rhythmicity is also apparent in the baseline condition. The explanation “even at baseline, firing shows some relationship to a (for example) 8-Hz signal” does not make much sense: there cannot be a relationship between a brain signal and “fictive light pulses”. Therefore a reasonable course of action is to compare the “apparent rhythmicity” in baseline to stimulation. There seem to be a substantial difference in the examples; group statistics on this would be a valuable addition.

Reviewer #2 (Remarks to the Author):

The revision has addressed my concerns sufficiently. I do recommend that authors to do use tetrodes in the future as place cells with non-overlapping place fields may not show strongly correlated firing. Therefore, even isolated units with clean refractory periods might contain the activity of more than one cell. Yet, the effort was sufficient in the revision to ensure that well-isolated units were included in any further analyses.

Response to Reviewers

Reviewers' comments:

Reviewer #1 (Remarks to the Author):

Overall, the authors were responsive to many of my points and the manuscript has significantly improved. I was especially impressed by the authors showing that the PV stimulation results are independent of septal inputs and by the analysis of ripples. However, I have some remaining concerns (see below in details). Beyond the methodological concern of cluster quality,

- I find that the results of PV inhibition on theta and delta oscillations are not entirely convincing.

- there is the lingering concern about the significance of optogenetic PV stimulation, as it was performed under anesthesia and without behavioral correlates.

We appreciate the opportunity to address these concerns, and are pleased that we were able to improve the manuscript. Regarding the latter point, the reviewer rightly points out the possibility that the effects of optogenetic stimulation on network dynamics might not be identical in anesthetized vs. non-anesthetized hippocampus. To address this, we implanted a ChR2-expressing mouse with a stereotrode array to record CA1 neuronal activity in a freely-behaving condition, during waking behavior, with and without the rhythmic stimulation used in our anesthetized recordings. Data from neurons and LFPs recorded under these conditions are now shown in **Supplementary Fig. 16**. As this figure shows, rhythmic (7 Hz) stimulation of CA1 PV+ interneurons is sufficient to entrain firing of principal neurons' firing, and induce rhythmic LFP activity, during waking behavior.

The numbers refer to my original points.

1. Still, a random check of papers from the Pubmed search the authors referred to showed that it is uncommon to study "fast-spiking" interneurons without any quantification of their firing rates. Identification procedures often include double criteria based on spike width and firing rate. So as a minimum, the firing rate distribution of "FS interneurons" in this study would be a valuable addition.

This is a fair point, and we thank the reviewer for pointing out that we had not provided these values. We now provide firing rate distributions for unit data for the two subclasses of neurons in revised **Supplementary Fig. 1B**. We also report mean firing rate values for the two cell populations, and statistics comparing them, in the figure legend. FS interneurons we identify in our studies do indeed fire at significantly higher rates than principal neurons.

Note that cluster quality does not only depend on separation of cluster pairs, but also separation of a putative single neuron cluster from everything else (including the so-called “noise cluster”). In that effect, Fig. S1c provides very limited information on cluster quality.

The metrics we provide (MANOVA and Davies-Bouldin [DB] validity index; **Supplementary Fig. 1c**) are standard in the field for assessment of spike cluster quality. Insofar as the major concern is single-unit spike separation (and avoidance of clusters with multiunit data), statistics comparing one spike data cluster with another spike data cluster in principal component space is reasonable. However, the reviewer raises a good point, in that separation from single unit data from background noise is an issue in some recording conditions. While inclusion of noise in principle component calculations during sorting is a confounding factor which we generally avoid, some investigators also include a “noise cluster” for statistical comparison with spike clusters, which we now also provide. Data for statistical comparisons of spike-vs.-noise are now shown in revised **Supplementary Fig. 1C**, to provide a more complete picture of data quality for these experiments. These comparisons are now side-by-side with our spike data-only comparisons. As shown in the figure, MANOVA p -value statistics for spike-vs.-noise compare very favorably with those from spike data-only, while DB validity indices shift to slightly higher values. The latter effect is likely explained by the facts that: 1) we had very low levels of background noise in these recordings and 2) the sparse noise that was present was not well-clustered in principal component space, compared with spike data. We have revised the figure legend and the **Methods** section to clarify the distinction between the two sets of measures.

The analysis of ripple in Fig.1f-g is a good addition.

We are grateful to the reviewer for suggesting additional analyses of these data, which we feel have improved the manuscript.

5. The representative LFP spectrograms do not really make up for the lack of thorough analysis I raised in my comment. In fact, they rather suggest that the mean effect could have been a result of rather sporadic events than typical oscillatory behavior, which diminishes my enthusiasm about this result.

The reviewer raises a good point. These increases in delta or theta power (or even occurrence of ripples) may wax and wane, rather than being constant. Indeed, we think that this is likely, given the fact that our FSM analysis suggests that certain network-wide activity patterns may be occurring intermittently over the course of several hours (as shown in **Fig. 7** and **Supplementary Figs. 10 and 11**). We now address this possibility in the revised **Results** section. However, it is unclear how such a result would affect interpretation of the present data, because the timescale of memory formation is on the course of hours, not seconds or minutes - presumably, changes in network dynamics over this longer timescale (encapsulated by average - not moment-to-moment - spectral power) are more salient for the process of memory consolidation.

It is also worth noting that the standard quantitative analysis in the sleep field (and in most fields dealing with LFP activity) is power spectral density *across* a given state or condition (e.g., NREM, REM, and wakefulness). There is simply no precedent for large-scale spectrogram analysis over the timescales (i.e., tens of hours) covered in this study. To carry out such an analysis (and to interpret the significance of the resulting data), while an interesting idea, is far beyond the scope of this manuscript.

A harmonic is always a positive integer multiplier of the base frequency, so theta harmonics appear above the theta range.

We are grateful to the reviewer for pointing out that we made a misstatement in our previous response. We were incorrect in our use of the term “harmonic”. The word that we should have used was “subharmonic”, referring to an oscillation that is an integer *sub*multiple of the base (theta) frequency - in this case, $f/2$. We apologize for causing confusion by substituting an incorrect term for the phenomenon we were attempting to explain. Because we very frequently see an $f/2$ peak in our LFP data, we suspect that the lower-frequency peak is a subharmonic. To demonstrate how robust this phenomenon is, we show at below the raw PSDs (bottom) for two LFPs (top) recorded from CA1 in a mouse receiving 7-Hz optogenetic stimulation. This stimulation pattern generates robust oscillations at 7 Hz, but also introduces peaks in the PSD at both 14 Hz (a $2f$ harmonic) and 3.5 Hz (an $f/2$ subharmonic). This illustration is showing the raw LFP data, but in fact a 3.5 Hz subharmonic can also be seen in the LFP signal when it is band-pass filtered (i.e., from 6-8 Hz). The biological basis for generating this subharmonic oscillation is not clear, but from a mathematical standpoint such a subharmonic could be generated if the cycle-to-cycle amplitude of the base frequency oscillation is variable. This seems to be the case in our data, as is clear from the LFP traces shown here.

This is strengthened by the answer to minor point 2 - it is still unclear why control-CNO differs from vehicle.

This is related to the point we made regarding subharmonic oscillations, above. The single case (which the reviewer points out) in which there is an apparent difference between Control-CNO and hM4Di-Vehicle conditions is for 0.5-4 Hz PSD changes *in REM sleep* (while there is an increase in 4-12 Hz power in both groups). Because theta is strongly increased in REM in both of these groups, one might expect a subharmonic increase (possibly in the 0.5-4 Hz band) in both groups. We do not know why there is not as strong an increase in the lower (likely subharmonic) band during REM in Control-CNO mice (as shown in the PSD data in **Supplementary Fig. 7**) as there is for hM4Di-Vehicle mice, but we do not feel that this detracts significantly from the main findings of the paper.

6. The comparison of AMD to CCG is a nice addition; however, Fig. S9a is quite unclear.

We are glad that the reviewer found this comparison helpful. We hope that this will help illustrate the advantages of our current metrics over the more traditional cross-correlation-based measurements of network communication.

We regret that our description of **Supplementary Fig. 9a** was unclear, making it difficult to interpret. We thank the reviewer for bringing this to our attention, and we have revised the figure legend to make interpretation of this part of the figure more straightforward.

8. I appreciate the demonstration of CA1-specific PV-stimulation. Nevertheless, this as superior experimental design compared to the problematic original one, therefore it should be presented in the main figure and not in supplement.

We are glad that the reviewer agrees that this additional dataset has improved the manuscript. We have revised both **Fig. 8** and **Supplementary Fig. 15**, so that data from virally-transduced animals are now shown in the main figures, and data from transgenic animals are now shown in the supplement.

minor point 7. When the same data is presented repeatedly in a rhythmic fashion - like in these raster plots - there is an artificial rhythmicity. This is very easy to demonstrate and indeed the authors themselves do the demonstration: rhythmicity is also apparent in the baseline condition. The explanation "even at baseline, firing shows some relationship to a (for example) 8-Hz signal" does not make much sense: there cannot be a relationship between a brain signal and "fictive light pulses". Therefore a reasonable course of action is to compare the "apparent rhythmicity" in baseline to stimulation. There seem to be a substantial difference in the examples; group statistics on this would be a valuable addition.

We feel that we did not communicate clearly why there is some apparent relationship between neuronal firing (and LFP signals) and "fictive" rhythmic light pulses. Clearly, it would be nonsensical for us to suggest a causal relationship between fictive pulses and neural signals; we did not intend to imply

this. The mathematical relationship we see, however, is not artificial. From a quantitative standpoint it can (and in fact *must*) exist. The reason is as follows. If a data set composed of multiple rhythmic elements (e.g., an LFP or neuronal spike train) is plotted modulo *any* rhythmic signal that is within the frequency range of the data (i.e., where there is *some* spectral power in an LFP's PSD), a relationship between the data set and the rhythmic signal will be apparent.

Thus, the reason that we report apparent rhythmicity with respect to an 8-Hz signal is that there is *some* 8-Hz component to both spike timing and LFP activities at baseline. For this same reason, there is some coherence of firing by our measures at baseline (and this would be true for a range of frequencies). Simply put, coherence is *never* zero for such signals, so long as the reference is within the frequency range of the signal. This is not to say that 8 Hz is the *dominant* frequency that neurons or fields lock on to. For this reason, presentation of the baseline data with regard to a fictive rhythmic signal makes for a more fair quantitative comparison of data between baseline and stimulation conditions.

Groups statistics on cell-by-cell coherence changes from baseline are shown in revised **Fig. 8c** and **Supplementary Fig. 15**. We have revised the figure legends to make this more clear.

Reviewer #2 (Remarks to the Author):

The revision has addressed my concerns sufficiently. I do recommend that authors to do use tetrodes in the future as place cells with non-overlapping place fields may not show strongly correlated firing. Therefore, even isolated units with clean refractory periods might contain the activity of more than one cell. Yet, the effort was sufficient in the revision to ensure that well-isolated units were included in any further analyses.

We are pleased that we could adequately address the reviewer's concerns regarding our present data. We are grateful for their helpful suggestions, which have significantly improved the manuscript.

Reviewers' Comments:

Reviewer #1 (Remarks to the Author):

The authors adequately addressed most of my concerns and the manuscript has improved.

I accept the authors' argument on the spectral analyses.

However, I would like to still note that I do not find the un-anesthetized PV stimulation experiment very conclusive. The post-stimulation lasting effects were not replicated, which I deem the key results of the stimulation experiment, specifically, Fig.8f and g. (The fact that rhythmic PV stimulation synchronizes the network while it lasts is neither novel nor surprising in itself.) Also, I consider a minimal number of two animals a fair requirement if the authors wish to maintain their strong conclusion that 'following learning, PV+ interneurons drive CA1 oscillations and reactivation of CA1 ensembles' (Abstract).

Response to Reviewers

Reviewers' comments:

Reviewer #1 (Remarks to the Author):

The authors adequately addressed most of my concerns and the manuscript has improved.

I accept the authors' argument on the spectral analyses.

However, I would like to still note that I do not find the un-anesthetized PV stimulation experiment very conclusive. The post-stimulation lasting effects were not replicated, which I deem the key results of the stimulation experiment, specifically, Fig.8f and g. (The fact that rhythmic PV stimulation synchronizes the network while it lasts is neither novel nor surprising in itself.) Also, I consider a minimal number of two animals a fair requirement if the authors wish to maintain their strong conclusion that 'following learning, PV+ interneurons drive CA1 oscillations and reactivation of CA1 ensembles' (Abstract).

We appreciate the opportunity to address this additional concern. The addition of data from a single, non-anesthetized animal was in response to our misunderstanding of the reviewer's prior comment: "there is the lingering concern about the significance of optogenetic PV stimulation, as it was performed under anesthesia". We interpreted this as a concern about the effects of rhythmic PV stimulation under anesthesia *per se* - e.g., whether rhythmic stimulation was capable of entraining hippocampal rhythms in an awake, behaving animal. Thus, our aim was simply to demonstrate that the effects were qualitatively the same in an awake mouse (hence the $n = 1$ demonstration).

It was not clear from the previous comments that the reviewer was asking whether longer-term effects of PV+ interneuron stimulation (*i.e.*, changes in network stability and connectivity) are recapitulated in non-anesthetized mice. To address this additional question, we now present data from $n = 3$ non-anesthetized mice expressing ChR2, and $n = 3$ non-anesthetized mice expressing GFP, which underwent 2 h of rhythmic light delivery. Analysis of neuronal functional connectivity for neurons stably recorded across 2-h baseline, stimulus, and 2-h post-stimulus recovery indicated that as is true in anesthetized animals, there are apparent long-lasting effects of 7-Hz stimulation of PV+ interneurons on the CA1 network. While there was a clear trend for increased network stability up to 2 h post-stimulation in ChR2-expressing mice (which was not seen in GFP-expressing mice), this did not reach statistical significance (data now shown in **Supplementary Fig. 16c**). However, there was a clear and statistically significant change in functional connectivity strength across the network for up to 2 h following stimulation following rhythmic activation of PV+ interneurons. These data are summarized in **Supplementary Fig. 16d**. Taken together, we conclude that recordings from non-anesthetized mice do indeed replicate the results from anesthetized mice. These additional data support our general conclusions regarding the role of rhythmic PV+ interneuron activity in coordinating communication between neurons over behaviorally-relevant timescales (*i.e.*, hours).